# Kir4.1 channels contribute to astrocyte CO$_2$/H$^+$-sensitivity and the drive to breathe
Colin M. Cleary [1], Jack L. Browning [2], Moritz Armbruster [3], Cleyton R. Sobrinho [1], Monica L. Strain[1], Sarvin Jahanbani[2], Jaseph Soto-Perez[1], Virginia E. Hawkins [4], Chris G. Dulla[3], Michelle L. Olsen [2] & Daniel K. Mulkey [1] ✉

Astrocytes in the retrotrapezoid nucleus (RTN) stimulate breathing in response to CO$_2$/H$^+$, however, it is not clear how these cells detect changes in CO$_2$/H$^+$. Considering Kir4.1/5.1 channels are CO$_2$/H$^+$-sensitive and important for several astrocyte-dependent processes, we consider Kir4.1/5.1 a leading candidate CO$_2$/H$^+$ sensor in RTN astrocytes. To address this, we show that RTN astrocytes express Kir4.1 and Kir5.1 transcripts. We also characterized respiratory function in astrocyte-specific inducible Kir4.1 knockout mice (Kir4.1 cKO); these mice breathe normally under room air conditions but show a blunted ventilatory response to high levels of CO$_2$, which could be partly rescued by viral mediated re-expression of Kir4.1 in RTN astrocytes. At the cellular level, astrocytes in slices from astrocyte-specific inducible Kir4.1 knockout mice are less responsive to CO$_2$/H$^+$ and show a diminished capacity for paracrine modulation of respiratory neurons. These results suggest Kir4.1/5.1 channels in RTN astrocytes contribute to respiratory behavior.

Respiratory chemoreception is the mechanism by which the brain regulates breathing in response to changes in tissue CO$_2$/H$^+$, a process required to maintain unconscious breathing. The retrotrapezoid nucleus (RTN) is an important site of respiratory chemoreception[1]. A subset of excitatory RTN neurons sense CO$_2$/H$^+$ by inhibition of a leak K$^+$ channel (TASK-2)[2] and activation of G-protein coupled receptor 4 (GPR4)[3] and relay this information to other levels of the respiratory circuit to stimulate breathing. Evidence suggests that astrocytes contribute to RTN chemoreception by providing a CO$_2$/H$^+$-dependent purinergic drive that activates RTN neurons directly[4] and indirectly by mediating local vasoconstriction which limits removal of tissue CO$_2$/H$^+$ by blood flow[5,6]. The importance of astrocyte chemoreception to normal respiratory function is underscored by emerging evidence that disruption of CO$_2$/H$^+$ detection by astrocytes may contribute to breathing problems associated with neurological diseases including Keratitis-Ichthyosis-Deafness (KID) Syndrome[7] and Rett syndrome[8,9]. Despite this physiological significance, the basis for how RTN astrocytes sense changes in CO$_2$/H$^+$ is unclear.

Putative mechanisms implicated in RTN astrocyte chemoreception include inhibition of Kir4.1/5.1 channels[10,11], activation of the sodium bicarbonate cotransporter (NBC)[12,13], reverse mode operation of the Na$^+$/Ca$^{2+}$ exchanger (NCX)[12], and CO$_2$-dependent gating of Cx26 hemichannels[14,15]. These CO$_2$/H$^+$ sensing mechanisms are thought to facilitate release of glial transmitters including ATP[4,16,17], prostaglandins[18] or D-serine[19]. Scattered evidence suggests that these gilal transmitters are released by Ca$^{2+}$-dependent exocytosis[20] or possibly calmodulin-dependent[21] or -independent[14,15,22] modulation of connexin hemichannels. More recently, experiments in mice with a conditional deletion of NBCe1 (the main isoform expressed by RTN astrocytes[12]) had negligible effect on the central chemoreflex[23]. As Kir5.1 does not form functional homomeric channels, but instead forms heteromeric pH sensitive (pH sensitivity in the physiology range 7.0–7.3) channels with Kir4.1[10,24,25], we focused our efforts on understanding contributions of astrocyte Kir4.1 channels to RTN chemoreception. As a result of Kir4.1/5.1 channel inhibition by acidification[10], astrocytes should depolarize and promote Ca$^{2+}$ influx by reversal of the NCX, and may contribute to elevated [K$^+$]$_o$; therefore, we hypothesize that astrocyte Kir4.1/5.1 channels contribute to RTN chemoreception, and, as such, are important contributors of the drive to breathe.

To address this, we first performed single cell RNA sequencing (scRNA-seq) to characterize molecular signatures of astrocytes in the RTN region and transcription targets of astrocyte depolarization and

[1]Department of Physiology and Neurobiology, University of Connecticut, Storrs, CT, USA. [2]School of Neuroscience and Genetics, Genomics and Computational Biology, Virginia Tech, Blacksburg, VA, USA. [3]Department of Neuroscience, Tufts University School of Medicine, Boston, MA, USA. [4]Department of Life Sciences, Manchester Metropolitan University, Manchester, UK. ✉e-mail: daniel.mulkey@uconn.edu

gliotransmission. We identified three discrete subsets of astrocytes, notably, each expressed relatively high levels of Kir4.1 and Kir5.1 when compared to other RTN cell clusters. To definitively test involvement of Kir4.1 channels in astrocyte chemoreception and control of breathing, we generated an inducible astrocyte-specific Kir4.1 knockout (Kir4.1 cKO) by crossing $Gfap^{CreERT2/+}$ mice with $Kir4.1^{f/f}$ and Ai14 mice. We found that Kir4.1 cKO mice breathe normally in room air but fail to increase respiratory output in response to $CO_2$. This central chemoreflex deficit was mimicked by viral-mediated deletion of astrocyte Kir4.1 channels in the RTN region. Furthermore, the chemoreceptor deficit exhibited by Kir4.1 cKO mice was partially rescued by re-expression of Kir4.1 only in RTN astrocytes. At the cellular level, the proportion of RTN astrocytes depolarized by $CO_2/H^+$ decreased in slices from Kir4.1 cKO mice as did the purinergic-dependent modulation of RTN neurons and vasculature. Together, these results demonstrate that Kir4.1 channels contribute to RTN astrocyte chemoreception and respiratory output.

## Results

### Transcriptional landscape of RTN astrocytes

To date, there have not been any single cell analyses of glial cell types in the RTN and to what extent they contain Kir4.1 ($Kcnj10$) or other physiologically significant inward-rectifying potassium channels including Kir5.1 ($Kncj16$), Kir4.2 ($Kcnj15$), and Kir7.1 ($Kcnj13$). Therefore, we sought to characterize the transcriptional landscape of RTN astrocytes through scRNA-seq with a focus on identifying putative channels responsible for astrocytic depolarization and/or gliotransmission. These experiments were performed in single cells isolated from 10-day old wild type C57BL/6J and 4-hydroxytamoxifen-injected control mice ($N = 16$ mice; 8 of each sex) (Fig. 1). Neurons are identified by expression of $Snap25$, $Tubb3$, $Elavl2$, $Syp$ which correspond to clusters 16−18 (Fig. 1a. Supplementary Fig. 1), and since these clusters have been analyzed separately[26,27] we excluded them from further analysis here. Non-neuronal cells, comprising roughly 90% of the dataset. Of these, astrocytes were segregated from other glial cells based on expression of astrocyte transcriptional markers including $Aqp4$, $Sox9$, $Slc1a3$, $Slc1a2$, $Atp1a2$, $Slc6a11$, and $Gja1$[28,29] (Fig. 1a, b). Astrocytes in this region could be further subclassified into two groups (clusters 8 and 9, which

make up roughly 23% of all barcoded cells) based on the common expression of glial fibrillary acidic protein ($Gfap$) and high-temperature requirement A1 ($Htra1$), a serine protease implicated in cleaving insulin growth factor (Igf)-binding proteins. Distinguishing genes between these astrocyte clusters include insulin-like growth factor (IGF) binding protein 2 ($Igfbp2$), a secreted inhibitor of IGF that is expressed by cluster 8, whereas cluster 9 preferentially express myocilin ($Myoc$), which is associated with astrocyte-vascular interaction and the glial limitans[30] (Supplementary Fig. 1). We also classified one group as astrocyte-like (cluster 10) based on mixed expression of astrocyte markers (Fig. 1a, b). For example, cluster 10 expressed similar levels of $Cst3$ and $Igfbp2$ to other astrocyte-clusters, but also had globally distinguishing genes including $Vim$ (vimentin), an intermediate filament that is canonically expressed in cerebral astrocytes, $Slc6a13$ (GAT-2), a major GABA transporter, and $Gjb2$ (Cx26), a $CO_2$-sensitive gap junction for which defects have been implicated in congenital sensorineural hearing loss including KID syndrome as well as glial dysregulation[31]. We also found that astrocyte clusters 8−9 show high expression of $Kcnj10$ (Kir4.1) and $Kcnj16$ (Kir5.1) transcript compared to all other clusters (Fig. 1b, Supplementary Fig. 2). We detected minimal levels of $Kcnj15$ (Kir4.2) in clusters 8-9 (Supplementary Fig. 2). This is notable because Kir4.2 can heteromerize with Kir5.1, thus the absence of this binding partner together with co-expression of Kir4.1 and Kir5.1 suggests heteromeric Kir4.1/5.1 channels are the main inward rectifying conductance in clusters 8−9. In sum, our results indicate that there are at least two molecularly discrete groups of astrocytes in the RTN that may sense $CO_2/H^+$ through Kir4.1/5.1 heteromeric channels.

### Astrocyte specific inducible Kir4.1 condition knockout mice

In order to further investigate the role of Kir4.1 in astrocytes, we generated astrocyte-specific inducible Kir4.1 knockout animals (Kir4.1 cKO) by crossing mice that express the Cre/ERT2 receptor under the hGFAP promoter with mice that express a floxed-stop TdTomato reporter construct (Ai14; JAX # 007914) and floxed Kir4.1 (Kir4.1 $^{f/f}$; JAX # 026826)[32]. Kir4.1 cKO mice received injections of 4-hydroxytamoxifen (pups) or tamoxifen (adults) for all cellular, molecular, and behavioral experimentation. Pups were injected with 0.2 mg/kg of 4-hydroxytamoxifen for three days, starting

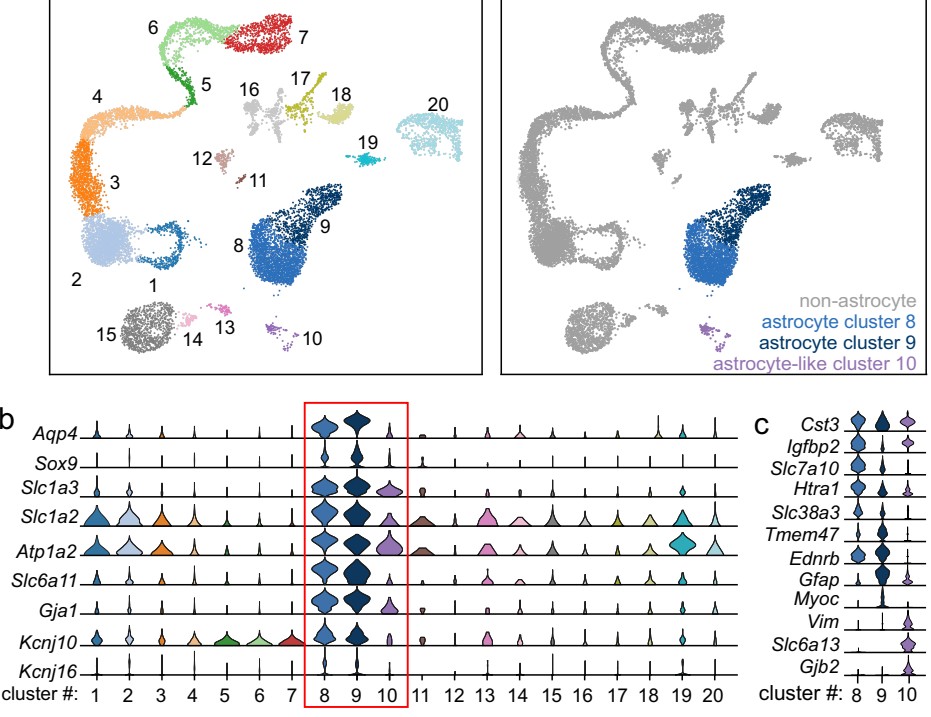

**Fig. 1 | scRNAseq of the RTN reveals molecular heterogeneity of astrocytes. a** clustering of all single cells from the RTN in postnatal day ten control animals represented in a tSNE plot; in total, 20 unique clusters emerged based on 1000 uniquely expressed genes and are denoted on left tSNE. Right-putative astrocyte clusters pseudocolored with all non-astrocytes grayed out. **b** Seven putative transcriptional markers for astrocytes compared across all clusters in violin plots, which identifies three putative astrocyte clusters (8−10) boxed in red. In addition, a comparison of $Kcnj10$ and $Kcnj16$ expression across all clusters identifies co-localization only in putative astrocyte clusters. **c** Subclustering analysis of clusters 8−10 reveal differential markers for each cluster. All violin plots span log based expression of 0 to 4 on the y axis.

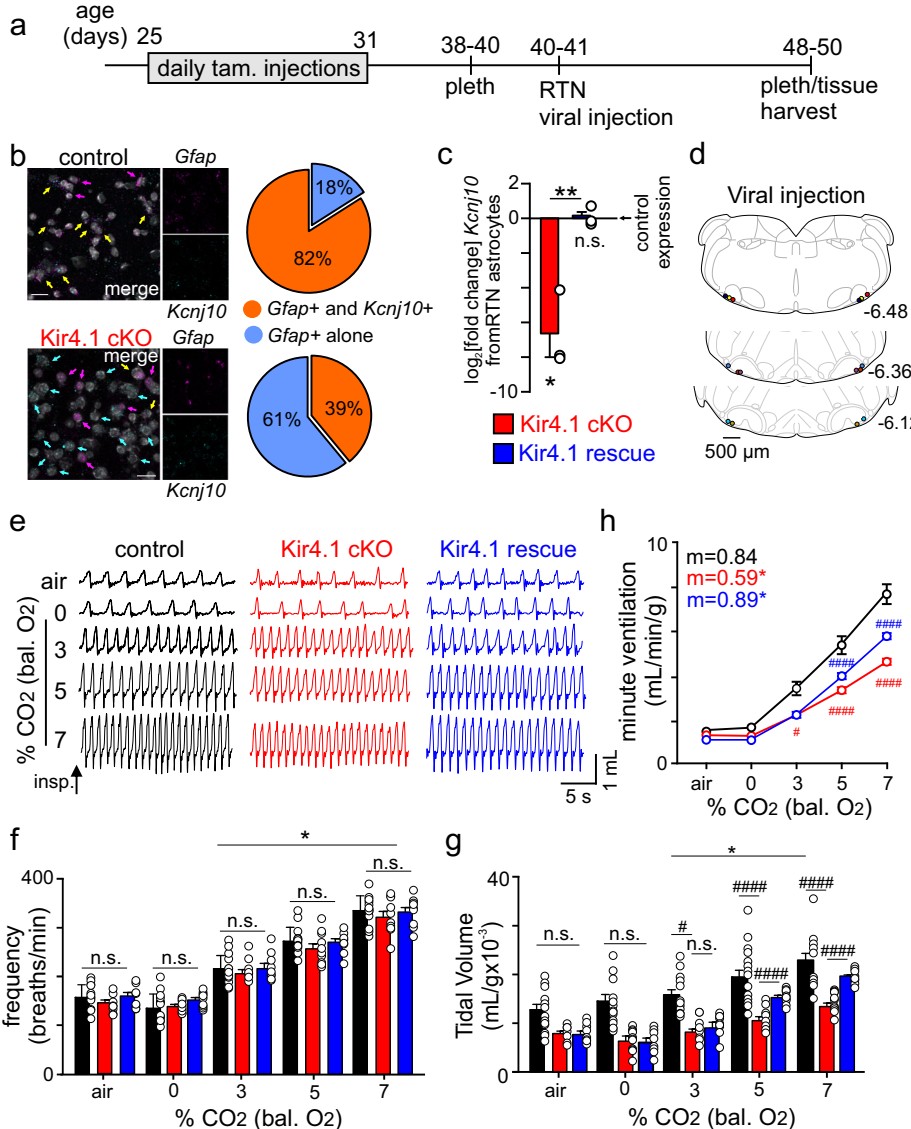

**Fig. 2 | Kir4.1 cKO mice have a reduced ventilatory response to $CO_2$, which is partially rescued by RTN specific re-expression of Kir4.1 channels. a** experimental timeline illustrating the tamoxifen regiment and timing of plethysmography experiments before and after RTN viral injections. **b** Left, images show fluorescent in situ hybridization labeling of *Kcnj10* (cyan puncta and arrows) and *Gfap* (magenta puncta and arrows), yellow arrows identify *Kcnj10* and *Gfap* co-labeled cells. Right, summary data ($N = 3$ animals/genotype) show that 82% of *Gfap* positive astrocytes in control tissue also express *Kcnj10* transcript. The proportion of *Gfap* and *Kcnj10* co-labeled cells was reduced to 39% in slices from Kir4.1 cKO mice. Scale bar 20 µm. **c** $\log_2$[fold change] values from *Kcnj10* (encodes Kir4.1) transcript amplification from enriched populations of RTN astrocytes from Kir4.1 cKO and Kir4.1 rescue tissue based on control expression. Astrocytes from Kir4.1 rescue mice show similar levels of Kcnj10 transcript expression to control ($T_2 = 0.3949$, $p > 0.05$). As expected, astrocytes from Kir4.1 cKO mice showed lower *Kcnj10* transcript expression compared to control ($T_2 = 5.178$, $p = 0.0353$) and Kir4.1 rescue tissue ($T_4 = 5.133$, $p = 0.0068$). **d** computer-assisted plots show the center of bilateral RTN injection of

AAV5-gfaABC1D-eGFP-Kir4.1 for Kir4.1 re-expression; each matching color pair of dots corresponds to one animal. Approximate millimeters behind bregma[52] are indicated by numbers next to each section. **e** traces of respiratory activity from control, Kir4.1 cKO and Kir4.1 rescue mice during exposure to room air and graded increases in $CO_2$ (0% to 7%; balance $O_2$), with corresponding summary data plotted as respiratory frequency (**f**), tidal volume (**g**) and minute ventilation (**h**). These data show that Kir4.1 cKO mice breathe normally under baseline conditions of room air and 100% $O_2$ but fail to increase respiratory output during exposure to $CO_2$ ($F_{1,14} = 21.28$, $p = 0.0004$). This central chemoreflex deficit is partly rescued by AAV5-gfaABC1D-eGFP-Kir4.1 mediated expression of Kir4.1 in RTN astrocytes ($F_{1,5} = 31.68$, $p = 0.0025$). Summary data are plotted mean and error bars are SEM. *, different from 0% $CO_2$; #, different between genotypes as assessed by two-way ANOVA with Tukey's multiple comparison test. ANCOVA used to compare linear regressions from 0 to 7% $CO_2$ (m = slope). One symbol $p < 0.05$; two symbols $p < 0.01$; three symbols $p < 0.001$; four symbols $p < 0.0001$.

at postnatal day three, followed by at least a 72-h recess period before experimentation. Adults were injected with 0.2 mg/kg of tamoxifen for seven days, starting at P25, followed by at least a seven-day recess period before experimentation. Mice that only express *Gfap*^CreERT2/+ or Kir4.1^f/f and the floxed-stop reporter construct were used as controls. We used fluorescent in situ hybridization to characterize *Kcnj10* transcript expression in *Gfap* expressing astrocytes in medullary sections containing the ventral parafacial region from control and Kir4.1 cKO pups. We found that 82% of

*Gfap*-positive cells in the RTN showed robust *Kcnj10* transcript expression, whereas only 39% of *Gfap*-positive cells in slices from Kir4.1 cKO mice express detectable *Kcnj10* transcript, which was expected given the incomplete nature of tamoxifen induced Kir4.1 knock-out in all astrocytes ($N = 3$ slices per animal, 3 animals per genotype, Fig. 2b). We also observed *Kcnj10* labeling in non-*Gfap* expressing cells that was invariant between genotypes. We suspect non-astrocytic *Kcnj10* labeling reflects known *Kcnj10* expression in the oligodendrocyte domain[33,34] (Fig. 1b) and so was

not considered a limitation of this model. Separately, we also tabulated the average number of *Gfap*+ cells in the RTN across genotypes; control slices had an average of 115 *Gfap*+ cells while Kir4.1 cKO slices had an average of 109 *Gfap*+ cells ($p > 0.05$). To further evaluate astrocyte specific *Kcnj10* expression in our model, we obtained an enriched population of RTN astrocytes from adult control and Kir4.1 cKO mice for qPCR analysis of *Kcnj10* transcript. We confirmed this population of astrocytes does not express neural (*Rbfox3*) or oligodendrocyte (*Olig1*) marker genes. Consistent with our in-situ results, we found that *Kcnj10* transcript was dramatically reduced in astrocytes from Kir4.1 cKO adult mice as compared to control astrocytes ($\log_2$[fold change] $3.1 \pm 0.8$; $T_4 = 3.101$, $p = 0.0362$; $N = 3$ adult animals per genotype, 3 technical replicates per gene assay) (Fig. 2c). These results demonstrate this to be a partial but incomplete Kir4.1 cKO model.

### Kir4.1 channels in RTN astrocytes contribute to the $CO_2/H^+$ dependent drive to breathe

Previous evidence suggests RTN astrocytes sense $CO_2/H^+$ by inhibition of Kir4.1/5.1[10,11] and release ATP to stimulate activity of RTN chemoreceptors directly[4,17,22] and indirectly by mediating vasoconstriction to limit tissue $CO_2/H^+$ removal by blood flow[5,6]. Therefore, we predict that if Kir4.1 channels are requisite determinants of RTN astrocyte chemoreception, then loss of Kir4.1 in this population would blunt the ventilatory response to $CO_2$. To test this possibility, we used whole-body plethysmography to measure baseline breathing and the ventilatory response to $CO_2$ in adult (38−40 days postnatal) control and Kir4.1 cKO mice. All mice received daily tamoxifen injections (0.2 mg/kg) for seven days starting at four weeks of age followed by a 1-week recess before any experimentation (Fig. 2a). We found that Kir4.1 cKO mice exhibit normal pattern and depth of breathing under baseline conditions of room air (21% $O_2$, balance $N_2$) and 100% $O_2$ (hyperoxia was used to suppress input from peripheral chemoreceptors) ($F_{1,27} = 0.0032$, $p > 0.05$; Fig. 2e−h; $N = 15$ control, 10 Kir4.1 cKO adult mice). We also noted no changes in apnea and sigh frequencies between control and Kir4.1 cKO mice in room air (apnea: $T_{10} = 0.6867$, $p > 0.05$; sighs: $T_{10} = 0.3780$, $p > 0.05$; $N = 6$ adult mice per genotype). Consistent with this, we found that Kir4.1 cKO mice also show normal arterial blood gases and baseline metabolic activity (Supplementary Fig. 3; $N = 5$ adult mice per genotype). However, consistent with our hypothesis, we also found that Kir4.1 cKO mice have a diminished capacity to increase respiratory output in response to $CO_2/H^+$ ($F_{4,56} = 22.2$, $p < 0.0001$). For example, minute ventilation in 7% $CO_2$ (balance $O_2$) for control and Kir4.1 cKO mice was $7.7 \pm 0.5$ ml/min/g ($n = 15$) and $4.6 \pm 0.1$ ml/min/g ($n = 15$), respectively ($F_{1,14} = 21.3$, $p = 0.0004$). This diminished $CO_2$-dependent drive to breathe was primarily mediated by a diminished tidal volume response to $CO_2$; a 7% increase in inspired $CO_2$ increased tidal volume by $23.0 \pm 1.5$ ml/ g in the control mice ($n = 15$) but only $14.2 \pm 0.6$ ml/g in Kir4.1 cKO mice ($n = 15$) (Fig. 2g) ($p = 0.0104$). These results show that global astrocyte loss of Kir4.1 disrupts central chemoreception.

To determine whether specifically RTN astrocytes contribute to this response, we used a viral approach to re-express Kir4.1 transcript in RTN astrocytes from Kir4.1 cKO mice (Fig. 2d−h; $N = 10$ Kir4.1 cKO adult mice) and separately delete Kir4.1 only in astrocytes in the RTN region in Kir4.1 f/f mice (Supplementary Fig. 4; $N = 6$ Kir4.1 f/f adult mice). To make RTN astrocyte specific Kir4.1 knockout mice, we injected AAV5-Gfap-eGFP-iCre (10 nL/side) bilaterally in the RTN of adult Kir4.1 f/f mice. Before and two weeks after surgery, we characterized breathing in the same animals with plethysmography. We found that RTN astrocyte-specific deletion of Kir4.1 resulted in a respiratory phenotype similar to global astrocyte Kir4.1 cKO mice. Specifically, these mice breathe normally under room air conditions but show a limited ability to increase respiratory output in response to $CO_2$ ($F_{1,5} = 31.68$; $p = 0.0025$) (Supplementary Fig. 4). In this case, the respiratory deficit involves diminished respiratory frequency ($F_{1,5} = 46.81$, $p = 0.001$) and tidal volume ($F_{1,5} = 12.45$, $p = 0.0168$) responses to $CO_2$. To establish the necessity of astrocyte Kir4.1 channels to RTN chemoreception, we made bilateral RTN injections of AAV5-gfaABC1D-eGFP-Kir4.1

(10 nL/side) to re-express Kir4.1 in global astrocyte Kir4.1 cKO mice (termed Kir4.1 rescue mice). To confirm this approached recovered *Kcnj10* to near normal levels in astrocytes specifically, we obtained enriched populations of RTN astrocytes from Kir4.1 cKO and Kir4.1 rescue mice (two weeks after intracranial injection). qPCR showed that Kir4.1 rescue mice had *Kcnj10* transcript levels ($\log_2$[fold change] $0.1 \pm 0.3$) similar to control ($T_2 = 0.3949$, $p > 0.05$) (Fig. 2c; $N = 3$ adult mice, 3 technical duplicates per gene assay). Importantly, we also found that re-expression of Kir4.1 in RTN astrocytes partly offset the adverse consequences of global astrocyte Kir4.1 channel deficiency. For example, consistent with loss of astrocyte Kir4.1 channels, Kir4.1 rescue mice also show normal respiratory activity in room air and 100% $O_2$ ($F_{1,9} = 0.001$, $p > 0.05$; Fig. 2h; $N = 10$ adult Kir4.1 rescue mice), suggesting these channels in astrocytes are dispensable for regulation of breathing under baseline conditions. We also found that Kir4.1 rescue mice show improved tidal volume responses to $CO_2$ ($F_{4,36} = 23.33$, $p < 0.0001$; Fig. 2g). We confirmed the specificity of AAV5-gfaABC1D-eGFP-Kir4.1 in RTN astrocytes as well as confirmed that bilateral RTN injections of control virus (AAV5-gfaABC1D.PI.Lck-GFP) had negligible effects on respiratory activity in wild type ($p > 0.05$) or Kir4.1 cKO mice ($p > 0.05$) (Supplementary Fig. 5; $N = 6$ adult mice/experiment). These results definitively establish astrocyte Kir4.1 channels as important determinants of RTN chemoreception and respiratory behavior.

### Mechanisms by which astrocyte Kir4.1 channels contribute to RTN chemoreception

Homomeric Kir4.1 channels and heteromeric Kir4.1/5.1 channels are important conduits for $K^+$ flux between intracellular and extracellular compartments and so contribute to maintenance of extracellular $K^+$ ($[K^+]_o$)[35]. However, Kir4.1/5.1 channels but not Kir4.1 homomers are strongly inhibited by acidification[24,25]; therefore, we reasoned that during high $CO_2/H^+$, heteromeric Kir4.1/5.1 channels do not contribute to $[K^+]_o$ buffering, thus allowing for the accumulation of $[K^+]_o$ to further augment $CO_2/H^+$-stimulated neural activity. To test this, we measured $[K^+]_o$ in the RTN region with an extracellular potassium electrode in slices from control animals during exposure to 10% $CO_2$. Consistent with expectations, exposure to $CO_2/H^+$ increased $[K^+]_o$ by $203 \pm 3.3$ μM (Fig. 3b, c; $N = 11$ slices). We also confirmed that RTN neurons show a modest excitatory response to increased $[K^+]_o$ between 300 and 500 μM (Fig. 3d, e; $N = 11$ cells). These findings are consistent across species, including rats (Fig. 3) and mice (Supplementary Fig. 6). Considering the ventral parafacial region also contains inhibitory neurons that limit activity of RTN neurons[26] and since high $[K^+]_o$ will likely stimulate activity of both excitatory and inhibitory neurons, its not clear how CO2/H + -induced increase in $[K^+]_o$ might impact respiratory drive. This is a novel mode of respiratory chemotransduction that warrants further investigation.

Astrocytes express a variety of electrogenic transporters that may influence neurotransmitter uptake or levels of intracellular $Ca^{2+}$. Therefore, we wanted to determine whether loss of Kir4.1 disrupts astrocyte voltage responses to $CO_2$. To test this, we used a viral delivery system to express a genetically encoded voltage indicator (Archon1) under control of the GFAP promotor as previously described[36]. Specifically, we made bilateral RTN injections of AAV5-Gfap-eGFP-Archon1 (250 nL/side) in adult control and Kir4.1 cKO mice ($N = 5$ mice/genotype). Eight weeks post-surgery, we confirmed that Archon1 expression is restricted to GFAP-labeled astrocytes (Fig. 4a top-left). For an experiment, brain slices were obtained from each genotype and Archon1 fluorescence was imaged in RTN astrocytes (soma and primary processes) during exposure to 10% $CO_2$. We found that ~75% ($N = 12$ of 16 cells) of RTN astrocytes in slices from control mice are depolarized by 10% $CO_2/H^+$ (depolarization is reflected as an increase in $\Delta F/F_0$), suggesting that the majority of RTN astrocytes are $CO_2/H^+$-sensitive, while a subset of RTN astrocytes are not directly responsible for respiratory chemoreception. However, we also found that control astrocyte $CO_2$-induced voltage responses are retained in the presence of $Ba^{2+}$ (1 mM) to block all Kir channels, desipramine (100 μM) to block Kir4.1 containing channels, and threo-beta-benzyloxyaspartate (TBOA; 100 μM) to block

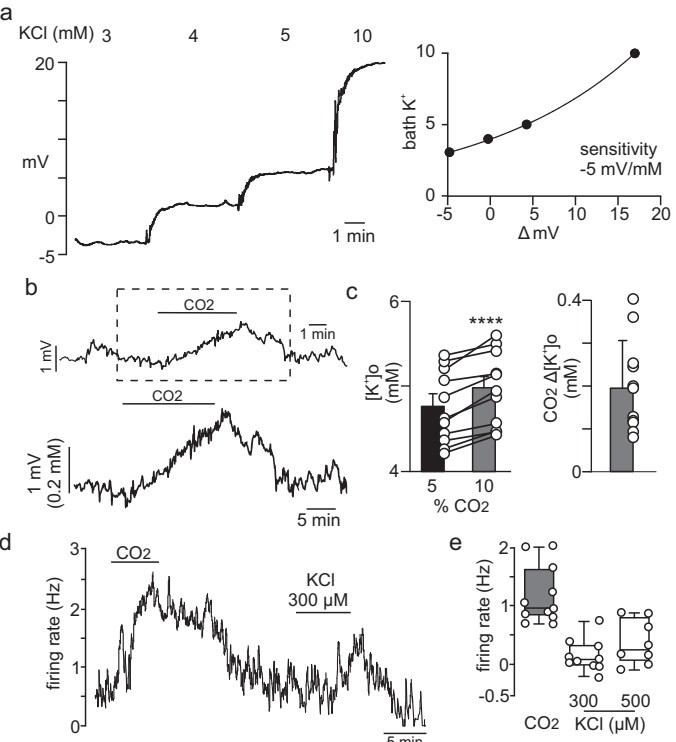

**Fig. 3 | Change in $[K^+]_o$ in the rat RTN as a result of bath acidification. a** trace of bath $[K^+]$ shows the stepwise application of $K^+$ to the perfusion media evokes a robust and stable change in electrode tip potential. Right, plot of mV change of the $K^+$ selective electrode in response to increases in concentration of $K^+$ (R2 = 0.997). **b** Traces of tip potential from a $K^+$ selective electrode positioned in the RTN region of a rat brainstem slice shows that exposure to 10% $CO_2$ causes a robust and reversible potential change indicative of increased $[K^+]_o$. The boxed in region in the top trace is replotted below on an expanded scale. **c** Summary data plotted as mean ± standard error shows the level of $[K^+]_o$ and $CO_2/H^+$ induced changes in $[K^+]_o$ (inset) in the RTN slices incubated in 5% and 10% $CO_2$ ($n = 12$, $p < 0.0001$, paired $t$ test). **d, e** Cell attached current clamp recordings were used to characterize the $[K^+]_o$ response of chemosensitive RTN neurons in slices from wild type rats. Note that RTN neurons were identified based on location and characteristic firing responses to $CO_2/H^+$ stimulus. Trace firing (**d**) and summary data plotted as mean with error bars as maximum/minimum values (**e**, $n = 8$ neurons) show that increasing extracellular $K^+$ increased RTN neural activity in a dose dependent manner.

glutamate transporters ($F_{3,25} = 0.9709$, $p > 0.05$) (Fig. 4a, c; $N = 9$ cells for $Ba^{2+}$, 8 cells for desipramine, 5 cells for TBOA). In contrast with these pharmacological results, we found that a smaller proportion of RTN astrocytes in slices from Kir4.1 cKO mice showed a voltage response to $CO_2$ (~50%; $N = 7$ of 13 cells) (Fig. 4b *bottom-right*) and those that did respond showed a smaller change in fluorescence compared to astrocytes from control tissue ($T_{17} = 2.161$, $p = 0.0453$) (Fig. 4b, c). These results suggest that removal of Kir4.1 channels in astrocytes precludes a majority of astrocytes from responding appropriately to increased levels of $CO_2/H^+$. Nonetheless, RTN astrocytes in slices from control rats[10] or Kir4.1 cKO mice (Fig. 4b, c) are depolarized by $CO_2/H^+$ even in the continued presence of $Ba^{2+}$ (1 mM), thus suggesting involvement of pH sensitive channels other than members of the Kir family. Overall, these results suggest $CO_2/H^+$-induced depolarization is a potential mechanism by which Kir4.1 contributes to RTN chemoreception. However, voltage-independent roles of Kir4.1 or non-Kir4.1 mediated mechanisms should also be considered.

## Loss of astrocytic Kir4.1 channels in the RTN alters purinergic-dependent chemoreception

Astrocytes in the ventral parafacial region provide a $CO_2/H^+$-dependent purinergic drive to stimulate activity of RTN neurons directly[17] and

indirectly by constricting local vasculature[5,6]; however, the role of Kir4.1 channels in these responses has not been tested. Therefore, we first characterized the $CO_2/H^+$-dependent purinergic modulation of RTN neurons in slices from control and Kir4.1 cKO mice. For these experiments, both genotypes received daily 4-hydroxytamoxifen injections (0.2 mg/kg) for three days starting at postnatal day three, followed by at least a 72-h recess before experimentation. In the cell-attached voltage-clamp mode, chemosensitive RTN neurons are identified based on location in the ventral parafacial region and their characteristic firing response to $CO_2$; spontaneous activity under control conditions (5% $CO_2$; pH 7.3) and at least a 1 Hz increase in activity in response to 10% $CO_2$ (pH 7.0). Compared to chemosensitive RTN neurons in control tissue, neurons from Kir4.1 cKO had higher basal activity ($T_{16} = 5.112$, $p = 0.0001$, $N = 10$ control cells, 6 Kir4.1 cKO cells; Fig. 5b). Contrary to previous evidence in rats[17], bath application of a pan-P2 purinergic blocker (PPADS; 100 μM) minimally affects basal activity of RTN chemoreceptors in slices from either genotype ($T_{13} = 0.0042$, $p > 0.05$). This finding suggests purinergic signaling does not contribute to integrated output of the RTN under baseline conditions in control or Kir4.1 cKO mice. This possibility is consistent with our in vivo evidence showing that both genotypes display similar respiratory activity under room air conditions (Fig. 2). We also found that PPADS (100 μM) decreased $CO_2/H^+$ sensitivity of RTN neurons in control tissue by ~32% ($T_8 = 3.936$, $p = 0.0043$) (Fig. 5b, c), whereas chemosensitive RTN neurons in slices from Kir4.1 cKO mice showed nearly identical $CO_2$ firing responses under control conditions (1.7 ± 0.3 Hz) and in PPADS (1.6 ± 0.2 Hz; $F_{2,10} = 0.2011$, $p > 0.05$) (Fig. 5d, e). These results suggest depletion of Kir4.1 from astrocytes disrupted purinergic modulation of RTN neurons.

We also tested the role of astrocytic Kir4.1 channels at the vascular level by characterizing $CO_2/H^+$-induced constriction of RTN arterioles in slices from control and Kir4.1 cKO adult brain slices. We confirmed that exposure to $CO_2/H^+$ (15% $CO_2$; pH 6.9) decreased the diameter of RTN arterioles in control tissue by $-4.6 ± 0.8\%$ ($T_9 = 4.335$, $p = 0.0019$, $N = 9$ vessels; Fig. 6a). This response could be blocked by pre-incubation (10 min) in PPADS (5 μM) ($T_9 = 1.263$, $p > 0.05$, $N = 9$ vessels) and mimicked by pharmacological activation of astrocytes (*trans*-ACPD; 50 μM) ($T_{10} = 4.077$, $p = 0.0028$, $N = 9$ vessels) or bath application of a P2Y$_2$ receptor agonist (PSB1114; 200 nM) ($T_7 = 3.032$, $p = 0.0191$, $N = 8$ vessels) (Fig. 6c). Based on evidence that RTN astrocytes also release prostaglandin (PGE$_2$) in response to $CO_2$[18,37], we also characterized $CO_2/H^+$ vascular reactivity in the presence of indomethacin (100 μM) to inhibit cyclooxygenase production of prostaglandins. We found that incubation in indomethacin (100 μM) did not alter $CO_2$-induced vasoconstriction of RTN arterioles in slices from control mice ($T_5 = 5.793$, $p = 0.0022$, $N = 6$ vessels; Fig. 6e). These results are consistent with previous evidence suggesting astrocytes and purinergic signaling contribute to $CO_2/H^+$ vascular reactivity of RTN arterioles[5,6]. RTN arterioles in slices from Kir4.1 cKO mice also constrict in response to $CO_2$ ($\Delta$ $-3.5 ± 1.1\%$; $T_9 = 3.728$, $p = 0.0047$, $N = 10$ vessels), *trans*-ACPD ($\Delta$ $-3.5 ± 1.1\%$; $T_{10} = 3.305$, $p = 0.0079$, $N = 10$ vessels) and PSB1114 ($\Delta$ $-3.0 ± 0.7\%$; $T_{10} = 3.419$, $p = 0.0066$, $N = 11$ vessels) (Fig. 6b, d). However, unlike control tissue, $CO_2/H^+$ vascular reactivity in Kir4.1 cKO slices was retained in the presence of PPADS ($\Delta$ $-2.3 ± 0.6\%$; $T_9 = 4.043$, $p = 0.0029$, $N = 10$ vessels) (Fig. 6b) and eliminated by indomethacin (100 μM) ($T_7 = 1.649$, $p > 0.05$, $N = 8$ vessels) (Fig. 6f). This is interesting because it suggests loss of astrocyte Kir4.1 disrupts $CO_2/H^+$ evoked ATP release while leaving prostaglandin in release unperturbed.

## Discussion

Astrocytes in the RTN contribute to respiratory drive by providing $CO_2/H^+$-dependent purinergic signals to regulate activity of respiratory neurons. Here, we establish that there are at least three molecularly distinct groups of astrocyte in the RTN that express Kir4.1 channels and may function as an important component of this response; loss of astrocyte Kir4.1 channels globally or specifically in RTN astrocytes blunted the ventilatory response to $CO_2$, whereas re-expression of Kir4.1 in RTN astrocytes improved the chemoreflex in Kir4.1 cKO mice. We also found that $CO_2/H^+$-dependent

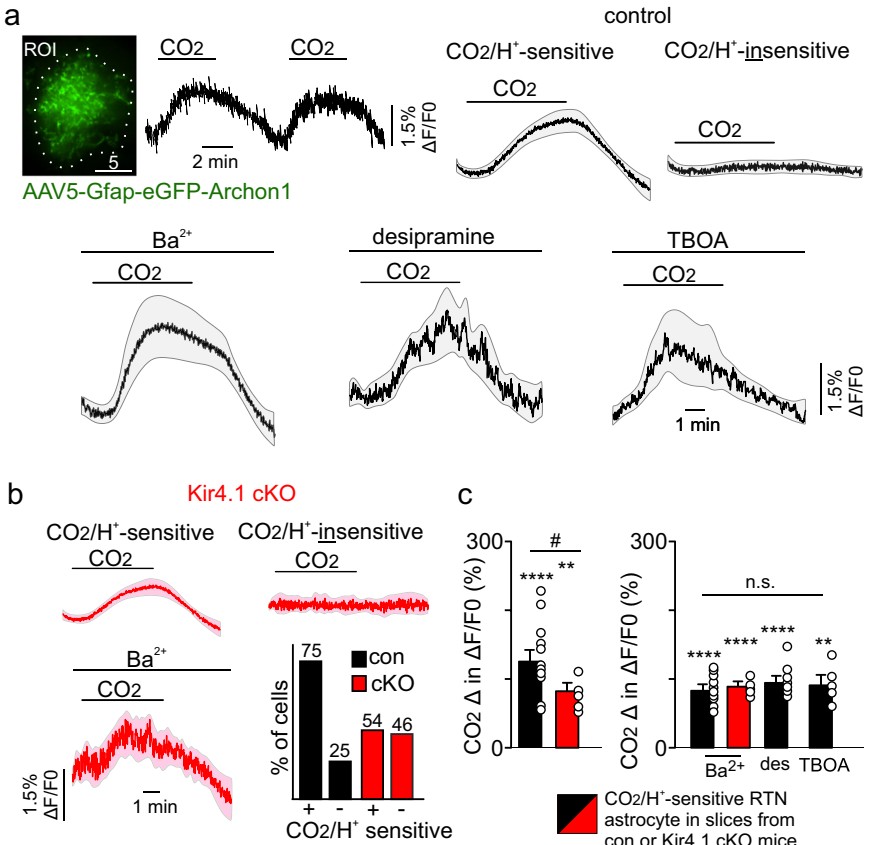

**Fig. 4 | CO$_2$/H$^+$ voltage responses of RTN astrocytes in slices from control and Kir4.1 cKO mice. a** top left: image of an RTN astrocyte transfected with AAV5-Gfap-eGFP-Archon1 and the corresponding change in fluorescent of that astrocyte in response to repeated exposure to 10% CO$_2$. Note that Archon1 is membrane bound and so predominantly reflects voltage responses in astrocyte processes[36]; therefore, astrocyte processes were included in the selected region of interest (dotted white line). Scale bar 5 μm. Top right and bottom: summary fluorescent traces from control astrocytes ($n = 16$ cells) show that most (75%) RTN astrocytes are depolarized in response to 10% CO$_2$/H$^+$ as evidenced by an increase in ΔF/F$_0$, whereas ~25% of RTN astrocytes did not show a ΔF/F$_0$ in response to CO$_2$/H$^+$. Traces of CO$_2$/H$^+$-induced ΔF/F$_0$ response of control RTN astrocytes was retained in the presence of Ba$^{2+}$ (1 mM) to block Kir channels, desipramine (100 μM) to block Kir4.1 channels and TBOA (100 μM) to block electrogenic glutamate transporters. **b** Summary fluorescent traces from Kir4.1 cKO astrocytes ($n = 13$ cells) show the amplitude and proportion of CO$_2$/H$^+$ responding astrocytes (54%) was reduced

compared to control. The CO$_2$/H$^+$-induced ΔF/F$_0$ response of RTN astrocytes in slices form Kir4.1 cKO mice was retained Ba$^{2+}$ (1 mM). These results suggest non-inward rectifying pH sensitive channels contribute to RTN astrocyte voltage responses to CO$_2$/H$^+$. Bottom-right: summary data showing proportion of CO$_2$/H$^+$-sensitive (i.e., depolarized) and -insensitive cells in each genotype. **c** Left, summary data plotted as mean ± standard error of peak ΔF/F$_0$ response shows that astrocytes in slices from control and Kir4.1 cKO mice respond to 10% CO$_2$; however, astrocytes from Kir4.1 cKO tissue show a smaller ΔF/F$_0$ response to CO$_2$ compared to RTN astrocytes from control tissue. Right, summary data shows CO$_2$/H$^+$-evoked ΔF/F$_0$ responses in Ba$^{2+}$, desipramine, and TBOA for astrocytes from each genotype. *, different from zero. #, different between genotypes. Responses to each condition are compared by one sample $t$ test, and responses between conditions by one-way ANOVA. One symbol = $p < 0.05$, two symbols = $p < 0.01$, four symbols = $p < 0.0001$. Gray and pink outlines for summary traces indicate SEM.

purinergic modulation of RTN neurons and arterioles was diminished in slices from Kir4.1 cKO mice, suggesting loss of Kir4.1 disrupts CO$_2$/H$^+$-evoked ATP release from RTN astrocytes. Although it remains unclear how Kir4.1 contributes to paracrine signaling by astrocytes, these results add to the limited list of potential cellular and molecular targets for treatment of hypoventilation-related breathing problems in disease.

Mechanisms by which RTN astrocytes sense and respond to CO$_2$/H$^+$ are not straightforward. Previous work suggests that CO$_2$-induced intracellular acidification activates the NBC to import HCO$_3^-$ to buffer pH$_i$ and Na$^+$ which is expected to favor Ca$^{+2}$ accumulation by reverse mode operation of the NCX and facilitate vesicular release of ATP[12]. In our scRNAseq dataset, we detected an abundance of NBCe1 transcript (*Slc4a4*) in all astrocyte clusters. However, recent evidence showed that astrocyte specific deletion of NBCe1 had negligible effects on baseline breathing or the ventilatory response to CO$_2$[23], thus suggesting the NBC is dispensable for astrocyte chemoreception. Conversely, CO$_2$/H$^+$ inhibition of Kir4.1/5.1 will depolarize astrocytes by decreasing the contribution of K+ efflux to the resting membrane potential (Goldman-Hodgkin-Katz voltage equation) and this is expected to favor reversal of the NCX (main NCX expressed by

astrocytes has a 3 Na$^+$:1Ca$^{2+}$ stoichiometry and under similar experimental conditions reverses near −80 mV[38]). In our scRNAseq dataset, we detected virtually no NCX (*Slc8a1*) expression across all three astrocyte subpopulations, so this possibility seems unlikely. It is also worth noting that CO$_2$-induced astrocyte depolarization was retained when Kir4.1 channels were pharmacologically blocked (Fig. 4b). This raises the interesting possibility that Kir4.1 channels contribute to RTN chemoreception at least in part by voltage-independent mechanism(s). Although understudied, ion channels can have enzymatic functions, and serve as adhesion molecules or integral components of a signaling complex[39]. Interestingly, evidence suggests that Kir4.1 is anchored by the dystoglycan complex with aquaporin 4 (*Aqp4*) which may function as CO$_2$ channels to increase CO$_2$ permeability[40]. Based on this, it is possible that loss of Kir4.1 results in a corresponding loss of Aqp4. In this case, it is possible that loss of Kir4.1 disrupts RTN astrocyte chemosensitivity by limiting intracellular pH responses to CO$_2$. Intriguingly, we have identified a subset of astrocytes with elevated levels of *Aqp4*, which seem to localize to the ventral surface. These astrocytes also express *Tmem47* and *Ednrb* (Fig. 1c), which are important for binding to epithelial cells and for maintenance of blood pressure through endothelin signaling.

**Fig. 5 | Purinergic signaling does not contribute to $CO_2/H^+$ sensitivity of RTN neurons in slices from Kir4.1 cKO mice. a** Example trace of firing rate (above) and segments of holding current (below) show the response of an RTN neuron from control tissue to repeated bouts of 10% $CO_2/H^+$ under control conditions and when ATP-purinergic receptors are blocked with PPADS. Initially, exposure to 10% $CO_2/H^+$ increased activity by ~2.3 Hz. A second exposure to $CO_2/H^+$ this time when P2 receptors are blocked with PPADS (100 μM) increases activity by ~1.8. $CO_2/H^+$ responsiveness was fully recovered after returning to control conditions for ~10 min. These results support the possibility that purinergic signaling contributes to $CO_2/H^+$ dependent output of the RTN in control animals. **b** Summary data shows that RTN neurons in slices from Kir4.1 cKO mice have higher baseline activity compared to control neurons ($T_{16} = 5.112$, $p = 0.0001$). This observation is consistent with the possibility that loss of Kir4.1 disrupts astrocyte $K^+$ buffering. **d** trace of firing rate (above) and segments of holding current (below) from an RTN neuron in tissue from a Kir4.1 cKO mouse shows high baseline activity and a vigorous response to $CO_2/H^+$ alone and in the presence of PPADS. **c, e** summary data plotted as $CO_2/H^+$ change in activity shows that PPADS blunted $CO_2/H^+$ sensitivity of control neurons by ~32% (**c**) but had negligible effect on $CO_2/H^+$ evoked activity of RTN neurons in slices form Kir4.1 cKO mice (**e**). Summary data are plotted as mean and error bars are standard error.

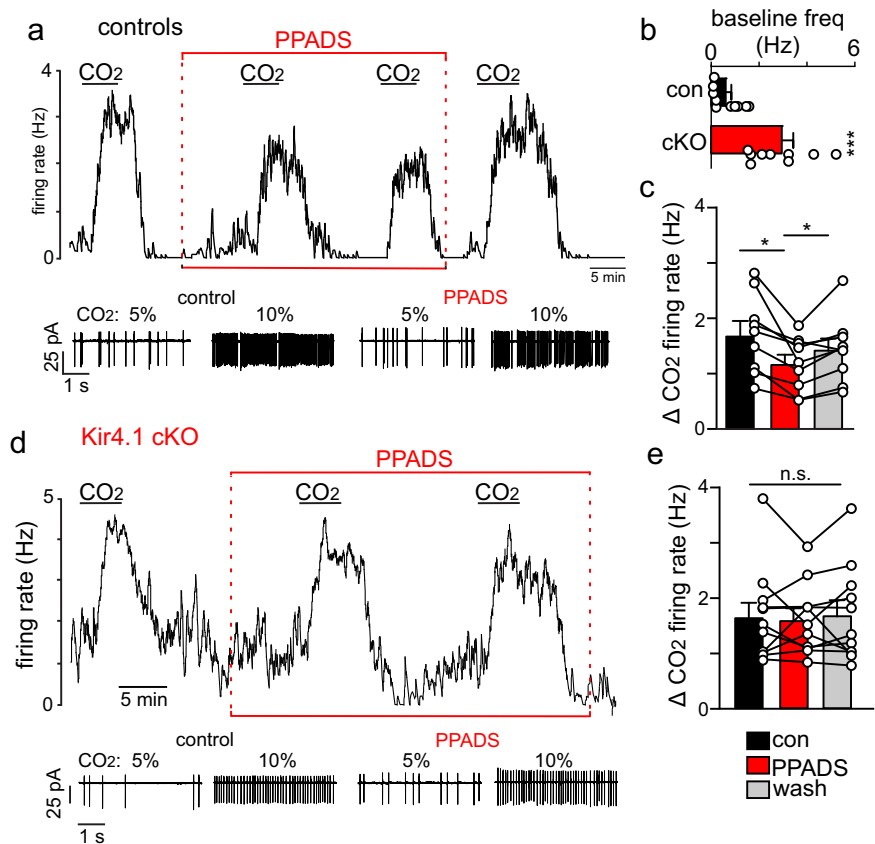

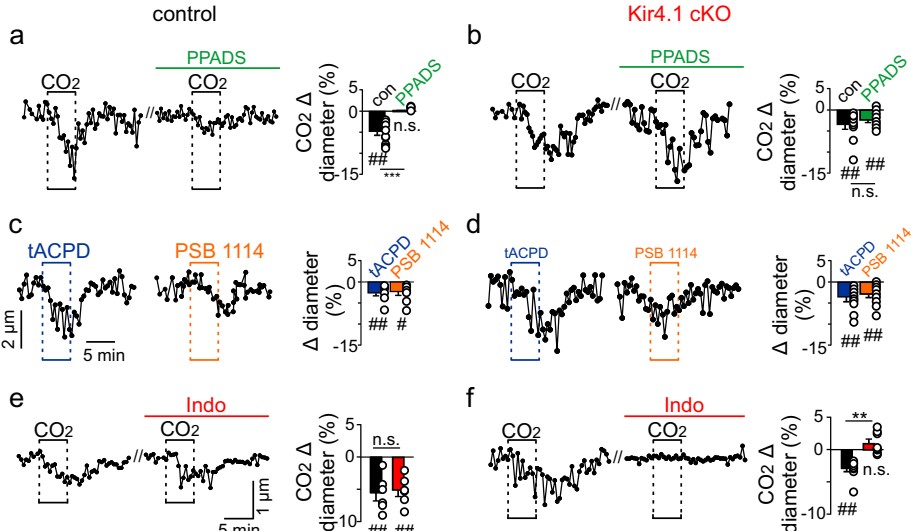

**Fig. 6 | Loss of astrocyte Kir4.1 channels decreases CO2/H + -dependent purinergic control of vascular tone in the RTN. a** traces of RTN arteriole diameter and corresponding summary data show that exposure to 15% $CO_2$ decreased diameter of vessels in slices from control ($T_9 = 4.043$, $p = 0.0029$) and Kir4.1 cKO ($T_9 = 3.728$, $p = 0.0047$) mice. However, mechanisms responsible for $CO_2/H^+$ vascular reactivity differ between genotypes. For example, incubation in PPADS (5 μM) to block ATP-purinergic signaling eliminated $CO_2/H^+$ vascular reactivity in control tissue (**a**) but minimally affected vascular responses in Kir4.1 cKO (**b**). **c, d** traces of RTN arteriole diameter and corresponding summary data show that $CO_2/H^+$-induced constriction was mimicked in each genotype by bath application of PSB1114 (100 μM) (control: $T_7 = 3.032$; $p = 0.0191$; Kir4.1 cKO: $T_{10} = 3.419$, $p = 0.0066$) or tACPD (50 μM)

(control: $T_{10} = 4.077$, $p = 0.0028$; Kir4.1 cKO: $T_{10} = 3.305$, $p = 0.0079$) to activate P2Y$_2$ receptors or astrocytes, respectively. **e, f** traces of vessel diameter and corresponding summary data show that $CO_2/H^+$-induced constriction was retained in RTN arterioles from control animals with bath application of indomethacin, a mixed COX-1/2 inhibitor that inhibits prostaglandin synthesis ($T_5 = 5.793$, $p = 0.0022$) (**e**) but eliminated in vessels from Kir4.1 cKO mice ($T_7 = 1.649$, $p > 0.05$) (**f**). summary data are plotted as mean with error bars showing standard error. [#], difference from baseline (*t* test). [*], differences in each condition (one-way ANOVA with Tukey's multiple comparison test). One symbol = $p < 0.05$, two symbols = $p < 0.01$, three symbols = $p < 0.001$, four symbols = $p < 0.0001$.

This suggests these astrocytes have close contact with blood vessels, thus making them well situated to sense changes in arterial $CO_2/H^+$.

Previous work showed that exposure to $CO_2$ elicited $Ca^{2+}$ signaling in RTN astrocytes that was proposed to facilitate vesicular release of ATP[4]. Conversely, prostaglandin release from RTN astrocytes is thought to occur through gap junction hemichannels[37]. Therefore, our evidence that loss of astrocyte Kir4.1 channels preferentially disrupted purinergic but not prostaglandin signaling suggests $CO_2/H^+$-evoked $Ca^{2+}$ responses are compromised in Kir4.1 cKO tissue. It should also be noted that $PGE_2$ is thought to facilitate $CO_2/H^+$-sensitivity of RTN neurons by activation of EP3 receptors[37]; however, since EP3 receptors are Gi coupled it is possible that retention of $PGE_2$ signaling in Kir4.1 cKO mice favors inhibition of RTN neural activity and diminished chemoreception. This possibility is yet to be determined.

Alternatively, the identification of elevated levels of *Slc38a3* (SNAT3) expression in parafacial astrocytes (Fig. 1c) may have a pH sensitive role in regulating neurotransmitter levels in the RTN. SNAT3 is a glycine transporter that involves the cotransport of $Na^+$ and glycine, and antiport of $H^+$. The transport of these molecules would be favored in acidic extracellular conditions, in which $H^+$ ions would be readily available, provided $Na^+$ is also available intracellularly[41]. Considering Phox2b neurons in the RTN are glutamatergic[26,42], the increased firing frequency in acidic conditions would require elevated levels of free glycine for the conversion to glutamate, which could be provided by this subset of astrocytes. Coupled with Kir4.1 inhibition via elevated $CO_2/H^+$, this could provide evidence for maintenance of elevated firing frequency of RTN neurons in an acidic environment.

The RTN as defined by expression of *Slc17a6*, *Phox2b*, *Nmb*, *Kcnk5*, *Gpr4*, and *Gal*[26,42] is an important site of chemoreception[1]. These glutamatergic neurons are intrinsically activated by $CO_2/H^+$ in vivo[43] and in vitro[44] and send excitatory projections to multiple levels of the respiratory network[45]. Selective activation of Phox2b neurons in the RTN caused a robust increase in breathing in both anesthetized[46] and awake[47] rats, whereas selective inhibition of RTN neurons decreased baseline breathing and the ventilatory response to $CO_2$ in awake[48] and anesthetized rats[49]. Perhaps the most compelling evidence RTN neurons are critical for chemoreception and do not require astrocytes to sense $CO_2/H^+$ was evidenced by the near complete elimination of the ventilatory response to $CO_2$ by conditional ablation of TASK2 channels and GPR4 receptors, which serve as the $H^+$ sensors RTN chemoreceptors[3]. These results strongly support the possibility that chemosensitive RTN neurons: (i) are excitatory; (ii) have the anatomical connectivity to regulate breathing; (iii) are intrinsically $CO_2/H^+$ sensitive; and (iv) are required for the ventilatory response to $CO_2$. However, loss of TASK2 and GPR4 from RTN neurons may also disrupt transmitter modulation of RTN neurons, thus it remains possible that $CO_2/H^+$-dependent output of the RTN and the chemoreflex and also subject to modulation by transmitters including ATP released by local astrocytes.

As described above, RTN astrocytes also function as chemoreceptors by sensing $CO_2/H^+$ and releasing ATP to activate RTN neurons to stimulate breathing. This response can be mimicked by pharmacological[11,50] or optogenetic activation of astrocytes[4]. However, contrary to results described above, there is evidence suggesting that $H^+$ sensitivity of RTN neurons was largely eliminated by pharmacological blockade of P2 receptors[4], suggesting neuronal $H^+$-sensitivity is dependent on purinergic drive from astrocytes. The most parsimonious interpretation of these divergent results is that astrocytes contribute in part to RTN chemoreception. Consistent with that, previous results show that blockade of P2 receptors at the level of the RTN decreased neuronal $CO_2/H^+$ sensitivity in acute brain slices[17] and the ventilatory response to $CO_2$ in awake[51] and anesthetized[17] rats by ~25%. Also, consistent with a chemo-modulatory role of astrocytes, we show here that disruption of astrocyte chemosensitivity by selective deletion of Kir4.1 channels from astrocytes blunted the ventilatory response to $CO_2$ (Fig. 2). However, the firing response of RTN neurons was otherwise retained in slices from Kir4.1 cKO mice, thus further confirming these cells are intrinsically $CO_2/H^+$ sensitive (Fig. 5).

In sum, our results show that Kir4.1 channels in RTN astrocytes are an important feature of respiratory chemoreception. Our ability to selectively disrupt Kir4.1 channels postnatally in astrocytes allowed us to unambiguously establish the extent to which Kir4.1 channels and astrocytes contribute to respiratory drive.

## Materials and methods
### Animals
All procedures were performed in accordance with National Institutes of Health and Animal Care and Use Guidelines from the University of Connecticut, Virginia Polytechnic Institute, Tufts University and Manchester Metropolitan University. Mice and rats were housed in a 12:12 light dark cycle with normal chow *ad libitum* if of weaning age. All mice, including *Gfap*[Cre/ERT2] (JAX # 012849), TdTomato (Ai14) reporter mice (JAX # 007914), and *Kir4.1* floxed (Kir4.1 [f/f]; JAX # 026826, gifted by Dr. Ken McCarthy from the Univ. North Carolina) were maintained on a C57BL6/J background. *Gfap*[Cre/ERT2]::Kir4.1 [f/f]::TdTomato[+/-] mice are defined at Kir4.1 cKO and control mice include *Gfap*[Cre/ERT2]::TdTomato[+/+] or Kir4.1 [f/f]::TdTomato[+/+] for experiments comparing controls to cKO animals. scRNAseq data and immunohistochemistry staining utilized C57BL6/J animals as aged-matched controls for experimentation. GFP+ Sprague-Dawley rats were also utilized only for extracellular $K^+$ recordings.

### Fluorescent in situ hybridization
Two-week-old control and Kir4.1 cKO mice, pre-injected with 4-hydroxytamoxifen for three days (0.2 mg/kg) starting at postnatal day three, were transcardially perfused with 20 mL of room temperature phosphate buffered saline (PBS, pH 7.4) followed by 20 mL of chilled 4% paraformaldehyde (pH 7.4) in 0.1 M phosphate buffered saline. The brainstem was then removed from the animal and post-hoc fixed for 24 h. After, the brainstem was rapidly frozen with dry ice and embedded with OCT compound. Alternatively, brains were freshly procured from pups and immediately frozen in a mixture of dry ice and 100% ethanol before sectioning. Brainstem slices (14 μm thick) containing the RTN were cryosectioned and collected onto SuperFrost Plus microscope slides and allowed to dry in the freezer for at least 1 h. Slices were then dehydrated with ethanol. This tissue was processed with the instruction of RNAscope Multiplex Fluorescent Assay (ACD, 320850); all probes used in our study were designed and validated by ACD and described in the figures and legends. Confocal images were obtained using a Leica SP8 and confocal image files containing image stacks were loaded into ImageJ for analysis.

### Unrestrained whole-body plethysmography
Respiratory activity was measured using a whole-body plethysmograph system (Data Scientific International; DSI), utilizing a small animal chamber maintained at room temperature and ventilated with room air or carbogen mixtures at a constant flow rate of 1.6 L/min. Before and after microinjection surgery, Kir4.1 control and Kir4.1 cKO mice were individually placed into a chamber and allowed to acclimate for 1 h prior to the start of an experiment. Respiratory activity was recorded using Ponemah 5.32 software (DSI) for a period of 20 min in room air followed by exposures to 0, 3, 5, and 7% $CO_2$ (balance $O_2$) (10 min/condition). Parameters of interests including respiratory frequency (FR, breaths per minute), tidal volume ($V_T$, measured in mL; normalized to body weight and corrected to account for chamber and animal temperature, humidity, and atmospheric pressure), and minute ventilation ($V_E$, mL/min/g) were measured during a 20 s period of relative quiescence, confirmed with synchronous video monitoring. All experiments were performed between 9 a.m. and 6 p.m. to minimize potential circadian effects.

### RTN viral injections
Adult Kir4.1 control and Kir4.1 cKO mice (>20 grams) were induced with 3% isoflurane. The right cheek of the animal was shaved, and an incision was made to expose the right marginal mandibular branch of the facial nerve. The animals were then placed in a stereotaxic frame and a bipolar stimulating electrode was placed directly adjacent to the nerve. Animals were maintained on 1.5% isoflurane for the remainder of the surgery. An incision

was made to expose the skull and two craniotomies (1.5 mm) were drilled left and right of the posterior fontanel, caudal of the lambdoidal suture. The facial nerve was stimulated using a bipolar stimulating electrode to evoke antidromic field potentials within the facial motor nucleus. In this way, the facial nucleus on the right side of the animal was mapped in the X, Y, and Z direction using a quartz recording electrode. One of the following viral vectors (AAV5-gfaABC1D-eGFP-Kir4.1, AAV5-Gfap-eGFP-iCre, AAV5-Gfap-eGFP-Archon1, or AAV5-gfaABC1D.PI.Lck-GFP) was loaded into a 1.2 mm internal diameter borosilicate glass pipette on a Nanoject III system (Drummond Scientific). One 10 nL injection of virus per side was delivered at least −0.02 mm ventral to the Z coordinates of the facial nucleus, to ensure injection into the RTN, with the exception of the AAV5-Gfap-eGFP-Archon1 virus, where 250 nL injections were made on each side of the brainstem to ensure adequate transfection of astrocytes. These same coordinates were used for the left side of the animal. In all mice, incisions were closed with nylon sutures and surgical cyanoacrylate adhesive. Mice were placed on a heated pad until consciousness was regained. Meloxicam (1.5 mg/kg) was administered 24 and 48 h postoperatively. Plethysmography was performed two weeks after viral injection, with the exception of the AAV5-Gfap-eGFP-Archon1 usage, which was incubated for a minimum of five weeks before slice experimentation. The location of all injection sites were later confirmed by *post hoc* histological analysis with the exception of the Archon1 experiments, where injection site did not affect the outcome of these experiments.

### Acute mouse pup brainstem slice preparation and in vitro electrophysiology

Slices containing the RTN were prepared as previously described[26]. In short, postnatal day 7−14 pups were anesthetized by administration of ketamine (375 mg/kg, I.P.) and xylazine (25 mg/kg; I.P.) and rapidly decapitated; brainstems were removed and transverse brainstem slices (250−300 μm) were cut using a microslicer (DSK 1500E; Dosaka) in ice-cold substituted Ringer solution containing the following (in mM): 260 sucrose, 3 KCl, 5 $MgCl_2$, 1 $CaCl_2$, 1.25 $NaH_2PO_4$, 26 $NaHCO_3$, 10 glucose, and 1 kynurenic acid. Slices were incubated for 30 min at 37 °C and subsequently at room temperature in a normal Ringer's solution containing (in mM): 130 NaCl, 3 KCl, 2 $MgCl_2$, 2 $CaCl_2$, 1.25 $NaH_2PO_4$, 26 $NaHCO_3$, and 10 glucose. Both substituted and normal Ringer's solutions were bubbled with 95% $O_2$ and 5% $CO_2$ (pH = 7.30).

**Electrophysiological recordings.** Individual slices containing the RTN were transferred to a recording chamber mounted on a fixed-stage microscope (Olympus BX5.1WI) and perfused continuously (~2 ml/min) with a bath solution of normal Ringer's solution (equilibrated with 5% $CO_2$; pH = 7.3). All recordings were made with an Axopatch 200B patch-clamp amplifier, digitized with a Digidata 1322 A A/D converter and recorded using pCLAMP 10.0 software. The firing response of chemosensitive RTN neurons to 10% $CO_2$ (duration of ~5 min) was assessed at room temperature (~22 °C) in the cell-attached voltage-clamp configuration (seal resistance >1 GΩ) with holding potential matched to resting membrane potential (Vhold = −60 mV) and with no current generated by the amplifier (Iamp = 0 pA). Patch electrodes had a resistance of 5−6 MΩ when coated with Sylgard 184 and filled with a pipette solution containing the following (in mM): 120 $KCH_3SO_4$, 4 NaCl, 1 $MgCl_2$, 0.5 $CaCl_2$, 10 HEPES, 10 EGTA, 3 Mg-ATP and 0.3 GTP-Tris, 0.2% Lucifer yellow (pH 7.30). Firing rate histograms were generated by integrating action potential discharge in 10 to 20 s bins using Spike 5.0 software.

### Acute rat pup brain slice preparation and in vitro [K + ]o electrode recording

Postnatal day 7−16 rat pups were anesthetized by $CO_2$ asphyxiation and rapidly decapitated; brainstems were removed and transverse brainstem slices (250−300 μm) were cut using a vibratome in ice-cold substituted Ringer solution containing the following (in mM): 3 KCl, 5 $MgCl_2$, 1 $CaCl_2$,

128 NaCl, 26.2 $NaHCO_3$, 10 glucose. Slices were incubated for 30 min at 37 °C and subsequently at room temperature in a normal Ringer's solution containing (in mM): 128 NaCl, 2 $MgCl_2$, 2 $CaCl_2$, 1.25 $NaH_2PO_4$, 26.2 $NaHCO_3$, and 10 glucose. Both substituted and normal Ringer's solutions were bubbled with 95% $O_2$ and 5% $CO_2$ (pH = 7.30).

**Fabrication of potassium sensitive microelectrodes.** Borosilicate glass capillaries (World Precision Instruments TW150F-4) were washed for 6 h with 1 M HCl and rinsed with 70% Ethanol, followed by incubation @ 120 °C overnight. Capillaries were then stored in an airtight container with anhydrous calcium sulfate desiccant until ready for pulling. Tips were created with a Narishige PC-10 pipette puller set to 84.8 and 61.2. Tips were then placed in a glass coplin jar and silanized by applying 50 μL of dichlorodimethylsilane (Tokyo Chemical Industry B2150) below the tips and sealed. The coplin jars were then placed in an oven @ 200 °C for 30 min. Following silanization, tips were visualized under a light microscope and tips manually broken to a size of ~5−10 μM. Tips were then loaded with 2 μL of potassium sensitive ionophore (World Precision Instruments IE190) and backfilled with KCl (100 mM). Microelectrodes were stored with tips down in a solution of KCl (100 mM) for up to ~1 h prior to use.

**Calibration and slice recordings with potassium sensitive microelectrodes.** Prior to slice recording, every microelectrode was calibrated, and calibration and bath solutions were made with either 10 mM glucose, 26.2 mM $NaHCO_3$, 2 mM $MgCl_2$, 2 mM $CaCl_2$, and either 3, 4, 5, or 10 mM KCl with 130, 129, 128, or 123 mM NaCl, respectively. Microelectrodes were then submerged in 4 mM KCl and a resistance test was completed in Clampex 10.7. Microelectrodes with a resistance outside the range of 100−1200 MΩ were not used for recordings. Current recordings were acquired using an AxonPatch 200 A amplifier (Axon Instruments) with a low-pass filter of 1 kHz and digitized on-line 10−20 kHz using a DigiData 1320 digitizing board. Data was acquired and stored using Clampex 10.7. The bath was perfused with 3, 4, 5, or 10 mM KCl for 3 min each to obtain voltage readings for each KCl concentration. Following calibration, RTN slices were placed into the bath with normal Ringer's solution bubbled with 95% $O_2$/5% $CO_2$, respectively, and visualized using a Zeiss Axio Examiner.D1. RTN was identified as the region between the facial nucleus and the ventral surface, and approximately 150−600 μM medial to the trigeminal nerve. Once the RTN was located, the electrode was placed into the RTN between the ventral surface and facial nucleus and allowed to equilibrate for 10 min. Following equilibration, the slice was perfused with Ringer's solution bubbled with 90% $O_2$/10% $CO_2$ for 5 min, and then Ringer's solution bubbled again with 95% $O_2$/5% $CO_2$.

After calibration and slice recording, the $\Delta[K^+]_o$ was determined by first generating a standard curve from the calibration recording based on the ΔmV from 4 to 3, 4 to 5, and 4 to 10 mM KCl. The corresponding standard curve was then used to compute the slice $[K^+]o$ by taking the ΔmV of the electrode out of slice, and the slice in Ringer's solution bubbled with either 5% or 10% $CO_2$ and plotting on the corresponding standard curve

### Acute adult mouse brainstem slice preparation
**Brainstem slice preparation.** Animals (P21 and older) were decapitated under isoflurane anesthesia and transverse brainstem slices (150 μm) were prepared using a vibratome in ice-cold substituted artificial cerebrospinal fluid (aCSF) solution containing (in mM): 130 NaCl, 3 KCl, 2 $MgCl_2$, 2 $CaCl_2$, 1.25 $NaH_2PO_4$, 26 $NaHCO_3$, 10 glucose. 0.4 mM L-ascorbic acid was added into aCSF while slicing. Slices were incubated for 30 min at 37 °C and subsequently at room temperature in aCSF, equilibrated with a 5% $CO_2$−95% $O_2$ gas mixture.

**In vitro Archon1 imaging and analysis.** Brainstem slices were placed in a submersion chamber on a custom Prior Open-Scope with X-light V2 spinning disk confocal microscope and a 60X water immersion lens, held in place with a harp and continuously perfused with aCSF,

equilibrated with 95% $O_2$/5% $CO_2$. An acceptable astrocyte for recording included sufficient eGFP signal as well as determinate resolution of soma and processes in greenfield on one plane. Once satisfying these criterion, an 11-min recording was initiated with both green (eGFP) and far-red (Archon1) channels including 1 min of baseline, 5 min exposure, and 5 min wash. After recording, raw image stacks were loaded into ImageJ and aligned. Following alignment, average $\Delta F/F_0$ images were generated and corrected for background fluorescence using individual stimulus conditions. The entire astrocyte was selected as a single ROI utilizing the eGFP channel and all far-red pixels within that ROI were averaged.

**In vitro arteriole imaging and analysis.** Arteriole imaging was conducted as previously described[5,6]. Prior to imaging, each slice was incubated for 35 min with 6 mg/mL DyLight 594 Isolectin B4 conjugate (Vector Labs) to label vascular endothelium. Individual brain slices containing either the RTN were transferred to a Plexiglas recording chamber mounted on a fixed-stage upright fluorescent microscope (Zeiss Axioskop FS) and perfused with 37 °C aCSF bubbled with a 5% $CO_2-21\%$ $O_2$ (balance $N_2$). Hypercapnic solutions were made by allowing aCSF to equilibrate with a 15% $CO_2-21\%$ $O_2$ (balance $N_2$). Arterioles were identified based on clear evidence of vasomotion under IR-DIC microscopy and bulky fluorescent labeling that indicates a thick layer of tightly wrapped smooth muscle surrounding the vessel lumen. All arterioles selected for experimentation had a luminal diameter between 8 and 50 μm. RTN vessels were located within 200 μm of the ventral surface and below the caudal end of the facial motor nucleus. For an experiment, fluorescent images were acquired at a rate of 1 frame every 20 s using a 40X water objective lens, a Clara CCD Andor camera, and the NIS Advanced Research software suite (Nikon). To induce a partially constricted state in arterioles, we continuously bath applied 125 nM of a thromboxane A2 receptor agonist (U46619). At the end of each experiment, we assessed arteriole viability by inducing a large vasoconstriction with a 60 mM $K^+$ solution and then a large vasodilation with a $Ca^{+2}$-free solution containing EGTA (5 mM), a phosphodiesterase inhibitor (papaverine, 200 mM), and an L-type $Ca^{+2}$ channel blocker (diltiazem, 50 mM). Only one vessel was recorded per experiment and slice. Any vessel that did not respond to these solutions were excluded in data analysis.

RTN vessel diameter was determined using ImageJ. All images were calibrated, and pixel distance was converted to millimeters. Data files underwent StackReg (Biomedical Imaging Group) to stack each image over time and then three linear region of interests (ROI) were drawn orthogonal to the vessel. A macro (available at https://github.com/omsai/blood-vessel-diameter) was used to determine peak to peak distance using fluorescence intensity profile plots for all slices of the data file.

**Single cell isolation**

Isolation protocols have been previously described[26]. In short, animals were euthanized under ketamine/xylazine anesthesia and brainstem slices were prepared using a vibratome in ice cold, high sucrose slicing solution containing (in mM): 87 NaCl, 75 sucrose, 25 glucose, 25 $NaHCO_3$, 1.25 $NaH_2PO_4$, 2.5 KCl, 7.5 $MgCl_2$, 0.5 mM $CaCl_2$ and 5 L-ascorbic acid. Slicing solution was equilibrated with a 5% $CO_2-95\%$ $O_2$ gas mixture before use. Transverse brainstem slices (300 μm thick) were prepared and then immediately enzymatically treated at 34 °C with protease XVIII (6 mg/mL, Sigma) for 6 min. After enzyme incubation, slices were washed three times in cold dissociation solution and then transferred to an enzyme inhibitor mix containing trypsin inhibitor (10 mg/mL, Sigma) and bovine serum albumin (BSA, 10 mg/mL, Sigma) in cold sucrose dissociation solution. Shortly thereafter, slices were transferred to a glass Petri dish on ice. Using a plastic transfer pipette and scalpel, each region of interest was micro-dissected out of the slices and manually separated into sterile micro-centrifuge tubes. Tissue chunks were then warmed to 34 °C for 10 min before trituration. A single cell suspension was achieved by trituration using a 25 and 30 gauge needles sequentially, attached to a 2 mL syringe. Samples

were triturated for an average of 5 minutes. Immediately after, the samples were placed back on ice and filtered through a 30-micron filter (Miltenyi Biotech) into sterile microcentrifuge tubes for cell viability assessment.

**Florescence-activated cell sorting (FACS) and pooled cell qRT-PCR.** FACS and qRT-PCR protocols have been previously described[26]. All cell types of interest were sorted on a BD FACSAria II Cell Sorter (UConn COR²E Facility, Storrs, CT) equipped with 407 nm, 488 nm, and 607 nm excitation lasers. Five minutes before sorting, 5 μL of 100 ng/mL DAPI was added to each sample. Cells were gated based on scatter (forward and side), for singlets, and for absence of DAPI. Cells were then gated to TdTomato or eGFP fluorescence and sorted by four-way purity into a sterile 96-well plate containing 5 μL of sterile PBS per sample (Supplementary Fig. 7). Between 100 and 500 cells were sorted per sample in any experiment and were processed immediately following FACS. A lysis reaction followed by reverse transcription was performed using the kit Taqman Gene Expression Cells-to-CT Kit (ThermoFisher) with 'Lysis Solution' followed by the 'Stop Solution' at room temperature, and then a reverse transcription with the 'RT Buffer', 'RT Enzyme Mix', and lysed RNA at 37 °C for an hour. Following reverse transcription, cDNA was pre-amplified by adding 2 μL of cDNA from each sample to 8 μL of preamp master mix [5 μL TaKaRa premix Taq polymerase (Clontech), 2.5 μL 0.2X Taqman pooled probe, 0.5 μL $H_2O$] and thermocycled at 95 °C for 3 min, 55 °C for 2 min, 72 °C for 2 min, then 95 °C for 15 s, 60°C for 2 min, 72 °C for 2 min for 16–20 cycles, and then a final 10 °C hold. Amplified cDNA was then diluted 2:100 in RNase-free $H_2O$. Each qPCR assay contained the following reagents: 0.5 μL 20X Taqman probe, 2.5 μL RNase-free $H_2O$, 5 μL Gene Expression Master Mix or Fast Advanced Master Mix (ThermoFisher), and 2 μL diluted pre-amplified cDNA. qPCR reactions were performed in triplicate for each Taqman assay of interest on a QuantStudio 3 Real Time PCR Machine (ThermoFisher). All three technical replicates were averaged to create one raw Ct values per Taqman assay. As a control, all samples were subject to cell markers of various cell types to confirm specificity: *Rbfox3* for neurons, *Aldh1l1* for astrocytes, and *Olig1* for oligodendrocytes. Any assay that did not give a discrete Ct value was given a Ct value of 40 for analysis.

**Single cell RNA sequencing**

A single cell suspension was made using the same protocol as described in the above section. Cell viability for each sample was assessed on a Countess II automated cell counter (ThermoFisher), and approximately 12,000 cells were loaded for capture onto an individual lane of a Chromium Controller (10X Genomics). Single cell capture, barcoding and library preparation were performed using the 10X Chromium platform according to the manufacturer's protocol (CG00052) using version 2 (v2) chemistry. cDNA and libraries were checked for quality on Agilent 4200 Tapestation, quantified by KAPA qPCR. All libraries were sequenced on individual lanes of an Illumina HiSeq4000 targeting 6000 barcoded cells with an average sequencing depth of 50,000 reads per cell.

**scRNA-seq data processing, quality control, and analysis.** Processing, qc, and analysis has been previously described for the same datasets shown in this publication[26]. In brief, Illumina base call files for both libraries were converted to FASTQs using bcl2fastq v2.18.0.12 (Illumina) and FASTQ files were aligned to the mm10 (GRCh38.84, 10X Genomics mm10 reference 2.1.0) using the version 2.2.0 Cell Ranger count pipeline (10X Genomics), resulting in two gene-by-cell digital count matrices. Downstream analysis was performed using Scanpy (v1.4.6). Individual libraries were subjected to quality control and filtering independently. Putative doublets were first removed using Scrublet on the raw matrixes. Then, for each matrix, cells containing fewer than 500 genes, more than 50 hemoglobin transcripts, or more than 30% mtRNA content were excluded from downstream analyses. Genes present in 5 or fewer cells, with fewer than 10 total counts were also excluded. For the dataset depicted in this work, we used three lanes of 10X Chromium chip with the

following conditions: P10 C57BL6/J pups, 4-OH Tamoxifen treated P10 C57BL6/J pups, and 4-OH Tamoxifen treated P10 Kir4.1 cKO pups. We used the first two lanes to create a comprehensive control database to compare Cre dependent and inducible lines with a C57BL6/J background strain. The individual filtered matrices (containing 3345 and 9054 cells respectively) were concatenated together resulting in an initial aggregated counts matrix of 12,399 cells by 15,923 genes. This aggregated count matrix was normalized by the total number of counts per cell then multiplied by the median number of counts across all cells, log2 transformed, and lastly scaled to zero mean and unit variance column-wise. For each matrix, cells containing fewer than 800 genes, more than 50 hemoglobin transcripts, or more than 20% mtRNA content were excluded from further analyses.

The 2000 most highly variable genes computed using 'scanpy.pp.highly_variable_genes' with 'flavor = "cell_ranger"' were selected for the computation of principal components (PCs). Genes related to cell cycle, stress response, the Y-chromosome, hemoglobin as well as ribosomal, mitochondrial, and the gene *Xist* were excluded from this list of highly variable genes prior to the computation of PCs. To further analyze the neuronal populations, all cells were first classified as neuronal or non-neuronal using a simple two-state Gaussian mixture model[26]. Non-neuronal cells were then identified based on the 1000 highest expressed genes and then sub-clustered for further analysis.

### Comprehensive lab animal monitoring (CLAMS)
Metabolic monitoring $O_2$ consumption ($VO_2$) and $CO_2$ production ($VCO_2$) was performed using comprehensive lab animal monitoring systems (CLAMS, Columbus Instruments). Adult mice were individually housed on a 12:12 light:dark cycle in plastic cages with a running wheel, regular bedding, and regular chow for one week before experimentation. Three days before the metabolic experiment, each animal was placed in the CLAMS housing cage with metered water and waste collection. Mice had two days to acclimate to the metabolic chamber; on the third day, all results were recorded for a continuous 24 h period (Oxymax v5.54). After data collection, all raw results were exported and averaged out per hour, only including times of no wheel activity as assessed by an activity monitoring system within the CLAMS housing cage. Then, light and dark periods were determined and averaged per animal for statistical analysis. Both sexes were equally represented in the data set.

### Blood gas analysis
Arterial blood gasses were collected from adult mice 6 weeks of age and older (>30 g) as previously reported[5,27]. In short, a RAPIDLab® 348 blood gas analyzer (Siemens) was used for all blood gas analysis; all calibrations, QC, and use were performed as indicated by the manufacturer. Animals were anesthetized with an induction dose of 3% isoflurane and then quickly switched to 1% isoflurane for the remainder of arterial blood collection. The left carotid artery was exposed and quickly cannulated to allow for arterial blood to be collected and analyzed by the blood gas analyzer; no more than 5 s was spent between blood collection and analysis on the blood gas analyzer.

### Statistics
Data are reported as mean ± standard error unless stated otherwise. Power analysis was used to determine sample size, all data sets were tested for normality using Shapiro-Wilk test, and outlier data points were identified by the Grubbs test and excluded from analysis. Statistical comparisons were made using t-test, Wilcoxon-ranked sum test, or one-way or two-way simple or repeated measures ANOVA or ANCOVA followed by Tukey's multiple comparison tests as appropriate. The specific test used for each comparison is reported in the figure legend and all relevant values used for statistical analysis are included in the results section.

### Reporting summary
Further information on research design is available in the Nature Portfolio Reporting Summary linked to this article.

## Data availablity
All numerical source data underlying all graphs in the paper can be found in supplementary data 1 file. Raw single cell RNA sequencing data have been deposited into GEO repository with accession codes GSE153172 GSE247704.

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

## Acknowledgements

This work was supported by the following National Institutes of Health Grants: R01HL104101 (D.K.M. and M.L.O.), R01HL137094 (DKM), R21NS134132 (D.K.M.) and F31HL167553 (M.S.), F31HL156470 (J.S-P), F31HL142227 (C.C.) and F32HL126381 V.H.). We thank Dr. Paul Robson, Dr. Elise Courtois, and Dr. William Flynn at Jackson Laboratories for their single cell and bioinformatics support.

## Author contributions

C.M.C.: Designed experiments, generated data, analyzed results, edited manuscript, approved final manuscript. J.L.B.: generated data, analyzed results, edited manuscript, approved final manuscript. M.A.: generated data, analyzed results, edited manuscript, approved final manuscript. C.R.S.: generated data, analyzed results, edited manuscript, approved final manuscript. M.L.S.: generated data, analyzed results, edited manuscript, approved final manuscript. S.J.: generated data, analyzed results, edited manuscript, approved final manuscript. J.S.-P.: generated data, analyzed results, edited manuscript, approved final manuscript. V.E.H.: generated data, analyzed results, edited manuscript, approved final manuscript. C.G.D.: Designed experiments, generated data, analyzed results, edited manuscript, approved final manuscript. M.L.O.: Designed experiments, generated data, analyzed results, edited manuscript, approved final manuscript. D.K.M.: Designed experiments, generated data, analyzed results, drafted manuscript, approved final manuscript

## Competing interests

The authors declare no competing interests.
