## [Peer Review File · Communications Biology]

Reviewers' comments:

Reviewer #1 (Remarks to the Author):

In the manuscript titled "Kir4.1 channels contribute to astrocyte CO₂/H⁺-sensitivity and the drive to breathe", the authors proposed a role of the Kir4.1/5.1 channels in RTN astrocytes contribute to respiratory behavior. Through single-cell RNA sequencing (scRNA-seq), the authors identify at least three molecularly distinct groups of astrocytes in the RTN expressing Kir4.1/5.1 channels. Subsequently, they employ astrocyte-specific inducible Kir4.1 knockout mice (Kir4.1 cKO) to investigate respiratory function, revealing that these mice exhibit normal breathing under room air conditions but display a diminished ventilatory response to CO₂. Notably, the chemoreflex impairment is partially alleviated by viral-mediated re-expression of Kir4.1 in RTN astrocytes. The study further highlights a reduction in CO₂/H⁺-dependent purinergic modulation of RTN neurons and arterioles in slices from Kir4.1 cKO mice. Molecular comparisons between control and Kir4.1 cKO samples unveil an increase in putative chemosensitive RTN neurons and a decrease in inhibitory neurons within Kir4.1 cKO tissue. Overall the findings are interesting and deserves publication in Communications Biology. However, some of the manuscript's conclusions lack robust support from experimental data. And the overall clarity of the study can be improved thorough discussion or elucidation of certain results.

1. While the anticipated consequence of Kir4.1 loss is an elevation in extracellular potassium ([K⁺]_o), leading to heightened neural activity and associated behaviors, the observed hypoventilatory phenotype in Kir4.1 cKO mice raises questions about inconsistency. The authors should make some efforts to provide an explanation.
2. The paper employs diverse species (rat and mouse) and spans various age groups (pups and adults) across different experimental procedures. For instance, pup mice are utilized in scRNA-seq, RNAscope, and electrophysiological recordings, whereas adults are involved in behavioral tests, in vitro Archon1 imaging, and in vitro arteriole imaging. The authors should provide a rationale for this diversity.
3. Discrepancies in GFAP expression between Fig2B and Fig7B require clarification. The control group shows more GFAP⁺ cells than the Kir4.1 cKO group in Fig2B, whereas an increased GFAP⁺ cells were observed in the Kir4.1 cKO by scRNA-seq in Fig7B. Please addressing this inconsistency.
4. Examining the relative distribution of chemosensitive RTN neurons through scRNA-seq elucidates a notable shift in the Kir4.1 conditional knockout (cKO) cohort when compared to controls. This analysis discloses an augmented proportion of excitatory neurons, presumably representing chemosensitive RTN neurons, accompanied by a corresponding decrease in the proportion of inhibitory neurons. To ensure the robustness of these findings, it may be necessary to validate the scRNA-seq data using complementary techniques such as RNAscope or immunostaining. Additionally, a comprehensive explanation is warranted to shed light on the underlying mechanisms driving the observed alterations in the balance between excitatory and inhibitory neurons within the RTN following Kir4.1 cKO.
5. In the RNAseq data of RTN (Fig1), it would be informative to explore changes in other Kir channels beyond those explicitly discussed in this paper (Kir4.1, Kir5.1, Kir4.2, and Kir7.1). Providing a comprehensive overview of all Kir channels in the RTN data would enhance the depth of the analysis.
6. The background disparity in the two immunofluorescence (IF) pictures from control and Kir4.1 cKO

mice in Fig2B raises questions about potential variations in staining conditions. Please clarify whether these images were stained at different times or under distinct conditions.

7. Please check and correct the spelling, e.g “Kir4.1” instead of “Kir.1” (line 179), “kir4.1 cKO mice” instead of “kir4.1 mice” (line 160).

Reviewer #2 (Remarks to the Author):

Cleary et al. address the expression of Kir4.1 channel as one possible sensory mechanism through which astrocytes in the RTN area can sense hypercapnia or acidosis. The topic is highly relevant for understanding the role of astrocytes in the central chemoreception, hence it is also important for understanding physiological and pathological events of respiratory disorders associated with central chemoreception deficits with a possible translational impact. The manuscript is well organized and written, and the results were obtained using several experimental approaches, which, in global terms, support the work hypothesis proposed by the researchers. Although there are puzzling results that will require further research to reach a better comprehension, the main hypothesis, that is, the proposal that astrocytic Kir4.1 channels contribute to central chemoreception at the RTN is well supported by the experiments. Such conclusion is original and up to my knowledge, non technical or conceptual flaws are found.

Main comments

This reviewer suggests to discuss: 1) Possible mechanisms that could explain the fact that conditional deletion of astrocytic Kir4.1 leads to genetic changes in other kind of cells and also affects the purinergic contribution; and 2) Since hypercapnia induces the closing of Kir4.1 channels, such reduction of potassium permeability per se depolarizes the astrocyte by decreasing potassium contribution to the resting membrane potential (Goldman-Hodgkin-Katz voltage equation).

In line 199 the authors say about mice with specific RTN astrocyte deletion of Kir4.1 and those with global deletion of Kir4.1: “these mice breathe normally under room air conditions but show a limited ability to increase respiratory output in response to CO₂”. However, in lines 209-213 they say that “re-expression of Kir4.1 in RTN astrocytes partly offset the adverse consequences of global astrocyte Kir4.1 channel deficiency. For example, consistent with loss of astrocyte Kir4.1 channels, Kir4.1 rescue mice also show normal respiratory activity in room air and 100% O₂”. I recommend rephrase these lines, because they are confusing and ascribing an improvement in the Kir4.1 rescue situation without a deleterious effect under baseline conditions of specific RTN astrocyte Kir4.1 KO.

Minor comments:

1) The authors perform a single cell analyses of astrocyte cell types in the RTN through scRNA-seq, and determined to what extent they contain Kir4.1 or other Kir channels. They described 3 clusters (8-10) and 1 possible cluster (11) of astrocytes. This is an interesting result, but based on figure 1 B and C, for this reviewer is not clear the criteria for separating astrocytes in different clusters since gene markers are expressed in all the astrocytes with some differences in the intensity of few genes. For example, those

clusters enumerated 8 and 9 looks very similar, except the intensity of Sox9 and Slc1a3. I suggest that authors explicit in the legend of these figures the criteria to classify as different each cluster of astrocytes.

2) In figure 2H, the ANCOVA was performed in the range 3 to 7% CO₂ or you use the values from air, 0, 3, 5 and 7%? Please indicate in the legend the range you used.

In this figure, what is the meaning of the parameter “m”. Please indicate in the legend.

3) Figure 3c, left graphic. I suggest to expand the [K] axis including only the range of individual values and join each individual value at 5% CO₂ with its respective 10% CO₂.

4) In lines 280-282, I lost the rational implied in the concatenation of the following phrases: “Contrary to previous evidence in rats, bath application of a pan-P2 purinergic blocker (PPADS; 100 μM) minimally affects basal activity of RTN chemoreceptors in slices from either genotype (T13=0.0042, p > 0.05). These results are consistent with in vivo evidence that Kir4.1 cKO mice breathe normally under baseline conditions (Fig. 2)”

5) In lines 778-779, the sentence: “however, astrocytes from Kir4.1 cKO tissue show a smaller ΔF/F₀ response to CO₂ compared to RTN astrocytes from control tissue” repeats information already mentioned in the line 776, and additionally, it appears out of the scope of the description of summary plots to the right.

6) Please be explicit respect to the timeline of manipulations for both, control and Kir4.1 KO treatment, previous to the study of genes in different cells of the RTN.

7) In line 382 replace “though” by “through”.

Reviewer #3 (Remarks to the Author):

The authors investigate the effect of the response to CO₂/H⁺ in the retrotrapezoid nuclei. To study astrocyte function is challenging, and especially in a small, deep nuclei as the RTN. The authors use available methods in an appropriate way and the controls are for the most part well chosen. The experiments are however technically challenging, and the effects measured are on the limit to be detectable. The authors need to show more of the original data to convincingly show that the effects are real. My major concern is thus how reliable the data is, and that more results need to be shown.

1. Figure 1 is not contributing to the results. Based on known astrocytic markers the authors identify 4 clusters of astrocytes, but the 4 clusters are not used for any further analysis. The authors mention a few genes that are highly translated, including Kir 4.1, but these are already known astrocytic genes and no new results come from this data.

2. The extracellular recordings of K⁺ are poorly described. What type of microelectrodes are used? How is the signal recorded and analyzed? The authors should show a larger part of the recording than just the peak.

3. Voltage differences in astrocytes are small and their function has been debated. The authors need to make sure that the signal that they see is specific (if a signal can be seen when cells are depolarized with high K⁺ for example). Is the signal sensitive to pH? Again, a larger part if the recording needs to be shown in the figure.

4. It is unclear what the top figures in 5a and d show? The figure legend says holding current, but the y axes say firing rate.

5. The claim that the numbers of interneurons decrease in Kir4.1 is remarkable and should be followed up with immunohistochemistry.
6. The number of samples is lacking in most case, and when n is reported it is not clear whether it refers to animals, slices or replications.

Reviewer #1 Comments:

Summary: *In the manuscript titled "Kir4.1 channels contribute to astrocyte CO₂/H⁺-sensitivity and the drive to breathe", the authors proposed a role of the Kir4.1/5.1 channels in RTN astrocytes contribute to respiratory behavior. Through single-cell RNA sequencing (scRNA-seq), the authors identify at least three molecularly distinct groups of astrocytes in the RTN expressing Kir4.1/5.1 channels. Subsequently, they employ astrocyte-specific inducible Kir4.1 knockout mice (Kir4.1 cKO) to investigate respiratory function, revealing that these mice exhibit normal breathing under room air conditions but display a diminished ventilatory response to CO₂. Notably, the chemoreflex impairment is partially alleviated by viral-mediated re-expression of Kir4.1 in RTN astrocytes. The study further highlights a reduction in CO₂/H⁺-dependent purinergic modulation of RTN neurons and arterioles in slices from Kir4.1 cKO mice. Molecular comparisons between control and Kir4.1 cKO samples unveil an increase in putative chemosensitive RTN neurons and a decrease in inhibitory neurons within Kir4.1 cKO tissue. Overall the findings are interesting and deserves publication in Communications Biology. However, some of the manuscript's conclusions lack robust support from experimental data. And the overall clarity of the study can be improved thorough discussion or elucidation of certain results.*

We thank the reviewer for their support and thoughtful suggestions.

Concerns:

1) While the anticipated consequence of Kir4.1 loss is an elevation in extracellular potassium ([K⁺]_o), leading to heightened neural activity and associated behaviors, the observed hypoventilatory phenotype in Kir4.1 cKO mice raises questions about inconsistency. The authors should make some efforts to provide an explanation.

As noted by the reviewer, loss of Kir4.1 will limit K⁺ uptake by astrocytes during stimulated neural activity and high [K⁺]_o is expected to further increase neural activity. However, the ventral parafacial region contains plenty of inhibitory neurons that limit activity of RTN neurons (PMID: 34013884) and since both neural populations are expected to respond similarly to high [K⁺]_o, there is no a priori reason to expect RTN dependent control of breathing to increase in Kir4.1 cKO mice. Furthermore, our results also show that loss of Kir4.1 blunted astrocyte voltage responses to CO₂/H⁺ and decreased purinergic modulation of RTN neurons. Loss of this purinergic drive to breathe is consistent with the hypoventilatory phenotype observed in Kir4.1 cKO mice. Therefore, in the discussion we speculated that altered paracrine signaling is the most likely cause of breathing problems in Kir4.1 cKO mice.

2) The paper employs diverse species (rat and mouse) and spans various age groups (pups and adults) across different experimental procedures. For instance, pup mice are utilized in scRNA-seq, RNAscope, and electrophysiological recordings, whereas adults are involved in behavioral tests, in vitro Archon1 imaging, and in vitro arteriole imaging. The authors should provide a rationale for this diversity.

Good point. The rationale for using neonatal animals in this work is twofold. First, the ventral parafacial region is highly myelinated and this makes it difficult to visualize cells for patch recording after ~2 weeks of age. Therefore, neonatal tissue was used for electrophysiological

experiments. Second, all previous transcriptomic analysis of the RTN was performed using neonatal tissue (PMID: 29066557; PMID: 34013884); therefore to facilitate comparison with previous work we chose to use a similar age for our astrocyte analysis. Regarding species, we chose to use rats for measurements of $[K^+]_o$ because the RTN region spans a larger area in rat compared to mice, thus allowing us to more accurately measure $[K^+]_o$ in the region of interest. Nonetheless, we confirmed that CO_2/H^+ induced changes in $[K^+]_o$ are similar in rats and mice (Suppl. Fig. 6), thus giving us confidence that these responses are retained across species.

In previous work, we confirmed that RTN neurons in slices from rat and mouse pups exhibit similar baseline electrical properties and firing responses to CO_2 and purinergic modulators (PMID: 30627640). We have also shown that RTN neurons in slices from rat pups respond to wake-on neurotransmitters (e.g., serotonin) in a manner similar to RTN neurons in awake or sedate adult rats (PMID: 23175845). Furthermore, CO_2/H^+ evoked voltage responses of astrocytes in slices from adult mice (Fig. 4) appear similar to our previous electrophysiological measurements in neonatal tissue (PMID: 20926613). Therefore, we believe fundamental mechanisms underlying regulation of RTN neurons and astrocytes are similar between species and across the ages used in this work. We have clarified these points in the text.

3) Discrepancies in GFAP expression between Fig2B and Fig7B require clarification. The control group shows more GFAP+ cells than the Kir4.1 cKO group in Fig2B, whereas an increased GFAP+ cells were observed in the Kir4.1 cKO by scRNA-seq in Fig7B. Please addressing this inconsistency.

To clarify, Fig. 2B shows the validation of our Kir4.1 cKO model. It shows the proportion of astrocytes that also express Kir4.1 transcript (minimum of five Kcnj10 labeled puncta colocalized with Gfap and DAPI) compared to the proportion of astrocytes that do not show Kcnj10 labeling. Importantly, these results show diminished expression of Kir4.1 transcript in Gfap+ cells in tissue from Kir4.1 cKO compared to control. We can confirm that the proportion of Gfap+ cells is similar between genotypes. We modified the text to make this point clearer.

4) Examining the relative distribution of chemosensitive RTN neurons through scRNA-seq elucidates a notable shift in the Kir4.1 conditional knockout (cKO) cohort when compared to controls. This analysis discloses an augmented proportion of excitatory neurons, presumably representing chemosensitive RTN neurons, accompanied by a corresponding decrease in the proportion of inhibitory neurons. To ensure the robustness of these findings, it may be necessary to validate the scRNA-seq data using complementary techniques such as RNAscope or immunostaining. Additionally, a comprehensive explanation is warranted to shed light on the underlying mechanisms driving the observed alterations in the balance between excitatory and inhibitory neurons within the RTN following Kir4.1 cKO.

We thank the reviewer for bringing this to our attention. We have decided to omit these results and corresponding Fig. because it is unclear at this time whether expression of cell type specific markers is affected by loss of Kir4.1 or proportions of excitatory and inhibitory neuron change in Kir4.1 cKO tissue. Also, speculation about this possibility would be premature.

5) In the RNAseq data of RTN (Fig1), it would be informative to explore changes in other Kir channels beyond those explicitly discussed in this paper (Kir4.1, Kir5.1, Kir4.2, and Kir7.1). Providing a comprehensive overview of all Kir channels in the RTN data would enhance the depth of the analysis.

As suggested, we expanded our analysis to include expression of all members of the Kir family of channels in clusters harvested from control tissue (new Suppl. Fig 2). We found that Kcnj10 (Kir4.1 transcript) and Kcnj16 (Kir5.1 transcript) are predominantly expressed by astrocyte clusters 8-9, whereas Kcnj15 (Kir4.2 transcript and Kir5.1 binding partner) is not detectable in these clusters, suggesting Kir5.1 mainly functions as a Kir4.1/5.1 heteromer in these cells. It is also worth noting that other members of the Kcnj family are not abundantly expressed by cells in this region at this developmental timepoint.

6. The background disparity in the two immunofluorescence (IF) pictures from control and Kir4.1 cKO mice in Fig2B raises questions about potential variations in staining conditions. Please clarify whether these images were stained at different times or under distinct conditions.

All images are taken under the same conditions at the same time and processes in the same batch. We have clarified this point in the methods. In any case, we agree that the image quality of the original version of this figure was suboptimal. Therefore, we generated new images that show more consistent fluorescence intensity and staining quality between genotypes (new Fig 2B).

7. Please check and correct the spelling, e.g. “Kir4.1” instead of “Kir.1” (line 179), “kir4.1 cKO mice” instead of “kir4.1 mice” (line 160).

Thanks. Fixed.

Reviewer #2 Comments:

Summary. Cleary et al. address the expression of Kir4.1 channel as one possible sensory mechanism through which astrocytes in the RTN area can sense hypercapnia or acidosis. The topic is highly relevant for understanding the role of astrocytes in the central chemoreception, hence it is also important for understanding physiological and pathological events of respiratory disorders associated with central chemoreception deficits with a possible translational impact. The manuscript is well organized and written, and the results were obtained using several experimental approaches, which, in global terms, support the work hypothesis proposed by the researchers. Although there are puzzling results that will require further research to reach a better comprehension, the main hypothesis, that is, the proposal that astrocytic Kir4.1 channels contribute to central chemoreception at the RTN is well supported by the experiments. Such conclusion is original and up to my knowledge, non technical or conceptual flaws are found.

We thank the reviewer for their support and helpful suggestions.

Main comments

1a) This reviewer suggests to discuss: 1) Possible mechanisms that could explain the fact that conditional deletion of astrocytic Kir4.1 leads to genetic changes in other kind of cells and also

affects the purinergic contribution

As noted above, we have decided to omit the sequencing results from Kir4.1 cKO mice because it is unclear whether expression of cell type specific markers is affected by loss of Kir4.1 or proportions of excitatory and inhibitory neuron change in Kir4.1 cKO tissue. Also, speculation about this possibility would be premature.

Potential mechanisms contributing to the observed shift in CO₂/H⁺ evoked astrocyte signaling from purinergic to prostaglandin was discussed in the text. Previous work suggests CO₂/H⁺ evoked Ca²⁺ in RTN astrocytes is important for ATP release. Conversely, CO₂/H⁺ evoked PGE₂ release is thought to involve gap junctions. Since expression of CO₂/H⁺-gated Cx26 increases in Kir4.1 cKO tissue, we believe gap junction signaling is favored in the Kir4.1 cKO model.

1b) Since hypercapnia induces the closing of Kir4.1 channels, such reduction of potassium permeability per se depolarizes the astrocyte by decreasing potassium contribution to the resting membrane potential (Goldman-Hodgkin-Katz voltage equation).

This statement has been added to the discussion.

2) In line 199 the authors say about mice with specific RTN astrocyte deletion of Kir4.1 and those with global deletion of Kir4.1: “these mice breathe normally under room air conditions but show a limited ability to increase respiratory output in response to CO₂”. However, in lines 209-213 they say that “re-expression of Kir4.1 in RTN astrocytes partly offset the adverse consequences of global astrocyte Kir4.1 channel deficiency. For example, consistent with loss of astrocyte Kir4.1 channels, Kir4.1 rescue mice also show normal respiratory activity in room air and 100% O₂”. I recommend rephrase these lines, because they are confusing and ascribing an improvement in the Kir4.1 rescue situation without a deleterious effect under baseline conditions of specific RTN astrocyte Kir4.1 KO.

We thank the reviewer for these comments. We have modified the text to clarify that there is derangement of MV at high CO₂ levels only and not in room air or 100% O₂ for either experiment.

Minor comments:

1) The authors perform a single cell analyses of astrocyte cell types in the RTN through scRNA-seq, and determined to what extent they contain Kir4.1 or other Kir channels. They described 3 clusters (8-10) and 1 possible cluster (11) of astrocytes. This is an interesting result, but based on figure 1 B and C, for this reviewer is not clear the criteria for separating astrocytes in different clusters since gene markers are expressed in all the astrocytes with some differences in the intensity of few genes. For example, those clusters enumerated 8 and 9 looks very similar, except the intensity of Sox9 and Slc1a3. I suggest that authors explicit in the legend of these figures the criteria to classify as different each cluster of astrocytes.

Thank you for bringing this to our attention. After further review/analysis of these data we decided to omit a cluster (former cluster 8) based a lack of globally distinguishing genes. All cells in our dataset are included based on containing >800 genes, <50 hemoglobin transcripts, and <20%

mtRNA content. Also as suggested, we explicitly report in the first section of the results the criteria used to define each astrocyte cluster.

2) *In figure 2H, the ANCOVA was performed in the range 3 to 7% CO₂ or you use the values from air, 0, 3, 5 and 7%? Please indicate in the legend the range you used. In this figure, what is the meaning of the parameter “m”. Please indicate in the legend.*

We performed ANCOVA from 0-7% CO₂. This was added into the figure legend for clarity. M= slope, this was added as a definition in the legend.

3) *Figure 3c, left graphic. I suggest to expand the [K] axis including only the range of individual values and join each individual value at 5% CO₂ with its respective 10% CO₂.*

Good suggestion. We have modified the figure as suggested.

4) *In lines 280-282, I lost the rationale implied in the concatenation of the following phrases: “Contrary to previous evidence in rats, bath application of a pan-P2 purinergic blocker (PPADS; 100 μM) minimally affects basal activity of RTN chemoreceptors in slices from either genotype (T13=0.0042, p > 0.05). These results are consistent with in vivo evidence that Kir4.1 cKO mice breathe normally under baseline conditions (Fig. 2)”*

Sorry for the confusion. Our evidence shows that bath application of PPADS did not affect baseline activity of RTN neurons in slices from control or Kir4.1 cKO mice, suggesting purinergic signaling does not contribute to integrated output of the RTN under baseline conditions in control or Kir4.1 cKO mice. This possibility is consistent with our *in vivo* evidence showing that both genotypes display similar respiratory activity under room air conditions (Fig. 2). We have clarified this section of the results.

5) *In lines 778-779, the sentence: “however, astrocytes from Kir4.1 cKO tissue show a smaller ΔF/F₀ response to CO₂ compared to RTN astrocytes from control tissue” repeats information already mentioned in the line 776, and additionally, it appears out of the scope of the description of summary plots to the right.*

Fig. 4B shows the CO₂ induced change in fluorescence over time for sensitive and insensitive astrocytes from each genotype. Similar data is plotted in Fig. 4C in this case plotted as peak CO₂ induced change in fluorescence for statistical analysis. We have modified the text to make these points more clear.

6) *Please be explicit respect to the timeline of manipulations for both, control and Kir4.1 KO treatment, previous to the study of genes in different cells of the RTN.*

A timeline is included to detect the *in vivo* experiments (Fig. 2A). Also, the first section of the results that describes the animal model describes the experimental design for all cellular and *in vivo* experiments. Specifically, we state “Kir4.1 cKO mice received injections of 4-hydroxytamoxifen (pups) or tamoxifen (adults) for all cellular, molecular, and behavioral experimentation. Pups were injected with 0.2 mg/kg of 4-hydroxytamoxifen for three days, starting

at postnatal day three, followed by at least a 72-hour recess period before experimentation. Adults were injected with 0.2 mg/kg of tamoxifen for seven days, starting at P25, followed by at least a seven-day recess period before experimentation”. In any case, we have modified the description of each set of experiments to make sure the experimental design is clear.

7) In line 382 replace “though” by “through”.

Corrected.

Reviewer #3 Comments:

The authors investigate the effect of the response to CO₂/H⁺ in the retrotrapezoid nuclei. To study astrocyte function is challenging, and especially in a small, deep nuclei as the RTN. The authors use available methods in an appropriate way and the controls are for the most part well chosen. The experiments are however technically challenging, and the effects measured are on the limit to be detectable. The authors need to show more of the original data to convincingly show that the effects are real. My major concern is thus how reliable the data is, and that more results need to be shown.

We thank the reviewer for their time and helpful suggestions. We appreciate the reviewers point that more could be done to increase confidence in the results. To that end, we have edited the text for clarity and made all suggested changes to figures. Also, as noted above we have omitted the sequencing data performed on cells isolated from Kir4.1 cKO because we are not able to differentiate changes in cell type specific genes from changes in proportions of cells.

1) Figure 1 is not contributing to the results. Based on known astrocytic markers the authors identify 4 clusters of astrocytes, but the 4 clusters are not used for any further analysis. The authors mention a few genes that are highly translated, including Kir 4.1, but these are already known astrocytic genes and no new results come from this data.

It is important to recognize that astrocytes are not a homogenous population with similar functions throughout the brain. At the level of the RTN, astrocytes in this region appear specialized to contribute to respiratory drive. For example, RTN astrocytes are unusually CO₂/H⁺ sensitive and to contribute to RTN chemoreception by paracrine release of ATP. Despite this functional significance, the molecular profile of RTN astrocytes is not known. Therefore, an important and novel aspect of this work is to define the molecular profile of RTN astrocytes. We also believe that including this information will facilitate future research targeted to each astrocyte subtype.

2) The extracellular recordings of K⁺ are poorly described. What type of microelectrodes are used? How is the signal recorded and analyzed? The authors should show a larger part of the recording than just the peak.

Thank you for bringing this to our attention. We have added additional information regarding the K⁺ electrode recordings to the methods. We have also added a longer example recording to Fig. 3.

3) Voltage differences in astrocytes are small and their function has been debated. The authors need to make sure that the signal that they see is specific (if a signal can be seen when cells are depolarized with high K^+ for example). Is the signal sensitive to pH? Again, a larger part of the recording needs to be shown in the figure.

We selected the Archon1 vector for its physiologic pH insensitivity, as the pKa has been demonstrated to be around pH 9-10 (PMID: 22120467, PMID: 36127376). In addition, GFP has a pKa around 6.0, which is also out of range for any physiologic or experimental acidification; as such, we believe both of these signals can be reliably used as voltage signals for our experimental paradigm. Nonetheless, we appreciate the concern of reproducibility. To address this, we modified the example in Fig. 4A to show a washable and repeatable $\Delta F/F_0$ response to 10% CO_2 .

4) It is unclear what the top figures in 5a and d show? The figure legend says holding current, but the y axes say firing rate.

We used cell attached voltage clamp mode to record neural activity over time; this is considered the preferred configuration for monitoring spontaneous neural activity (PMID: 16554092). The examples in Fig. 5 show firing rate (Hz) top and segments of holding current (pA) bottom. All labeling is correct. We have modified the legend to make this clearer.

5) The claim that the numbers of interneurons decrease in Kir4.1 is remarkable and should be followed up with immunohistochemistry.

As noted above, we have omitted these results.

6) The number of samples is lacking in most case, and when n is reported it is not clear whether it refers to animals, slices or replications.

We have added these details to the results and legends.

REVIEWERS' COMMENTS:

Reviewer #1 (Remarks to the Author):

The authors have sufficiently addressed my comments and I am happy to recommend publication of this paper.

Reviewer #2 (Remarks to the Author):

The authors have addressed all my concerns satisfactorily.

Reviewer #3 (Remarks to the Author):

Some of my comments have been handled, and the manuscript have benefited from increased clarity in the methods. The manuscript contains a considerable amount of results using different methods. My major concern however remains. Voltage responses in astrocytes are interesting and the experiments that the authors conduct are very promising. Voltage responses in astrocytes are hard to measure, and the differences that the authors report are very small. The traces shown are not enough to judge the quality of the recordings and it is questionable if these responses are really contributing to the effect (which the authors also acknowledge). The manuscript is however solid enough without the voltage measurements.

Reviewer #1 Comments:

The authors have sufficiently addressed my comments and I am happy to recommend publication of this paper.

Thanks again for your helpful suggestions and support!

Reviewer #2 Comments:

The authors have addressed all my concerns satisfactorily

Thanks again for your helpful suggestions and support!

Reviewer #3 Comments:

Some of my comments have been handled, and the manuscript have benefited from increased clarity in the methods. The manuscript contains a considerable amount of results using different methods. My major concern however remains. Voltage responses in astrocytes are interesting and the experiments that the authors conduct are very promising. Voltage responses in astrocytes are hard to measure, and the differences that the authors report are very small. The traces shown are not enough to judge the quality of the recordings and it is questionable if these responses are really contributing to the effect (which the authors also acknowledge). The manuscript is however solid enough without the voltage measurements.

We appreciate the reviewers feedback. We agree that astrocyte voltage responses are technically challenging to measure. For this reason, in this work we use the most cutting-edge approach available – genetically encoded membrane voltage indicator (Archon1) – and the only approach able to assay voltage in astrocyte processes where Kir4.1 channels are thought to be localized. Also, in previous work we confirmed that Archon1 shows robust signal-to-noise characteristics and is not sensitive to pH (PMID: 35484406). Therefore, we have high confidence in the quality of our measurements. Perhaps most importantly, using this tool we found that exposure to CO₂/H⁺ elicited statistically significant voltage responses that are fully washable and repeatable (Fig. 4). As such, these responses cannot be dismissed as artifacts.

As discussed in the text, the relevance of CO₂/H⁺-evoked astrocyte voltage responses remains unclear. Nevertheless, it is interesting that only a subset of astrocytes show CO₂/H⁺ voltage responses. This is consistent with previous slice-patch electrophysiology data (PMID: 20926613), and to the extent CO₂/H⁺ voltage responses reflect a physiological response (the relevance of which is yet to be determined), these results provide some of the only support in the literature for the possibility that astrocytes are functionally heterogenous. Therefore, we expect these results will be of broad interest to the field of astrocyte physiology and so we believe it is important to include these results.